# Principles of self-organization and load adaptation by the actin cytoskeleton during clathrin-mediated endocytosis

Matthew Akamatsu[1], Ritvik Vasan[2], Daniel Serwas[1], Michael A Ferrin[1], Padmini Rangamani[2]*, David G Drubin[1]*

[1]Department of Molecular and Cell Biology, University of California, Berkeley, Berkeley, United States; [2]Department of Mechanical and Aerospace Engineering, University of California, San Diego, La Jolla, United States

**Abstract** Force generation by actin assembly shapes cellular membranes. An experimentally constrained multiscale model shows that a minimal branched actin network is sufficient to internalize endocytic pits against membrane tension. Around 200 activated Arp2/3 complexes are required for robust internalization. A newly developed molecule-counting method determined that ~200 Arp2/3 complexes assemble at sites of clathrin-mediated endocytosis in human cells. Simulations predict that actin self-organizes into a radial branched array with growing ends oriented toward the base of the pit. Long actin filaments bend between attachment sites in the coat and the base of the pit. Elastic energy stored in bent filaments, whose presence was confirmed by cryo-electron tomography, contributes to endocytic internalization. Elevated membrane tension directs more growing filaments toward the base of the pit, increasing actin nucleation and bending for increased force production. Thus, spatially constrained actin filament assembly utilizes an adaptive mechanism enabling endocytosis under varying physical constraints.

*For correspondence:
padmini.rangamani@eng.ucsd.edu (PR);
drubin@berkeley.edu (DGD)

**Competing interests:** The authors declare that no competing interests exist.

## Introduction

Cells polymerize actin filaments to produce force and provide mechanical integrity for a variety of cellular processes, from cytokinesis and cell migration, to membrane reshaping and trafficking (*Pollard, 2016*). For each cellular process, actin filaments organize into a specific geometry that confers structural integrity and force-generation capacity. Most membrane deformation processes use branched actin networks nucleated by the Arp2/3 complex, a branched actin filament network nucleator (*Carlsson, 2018*; *Rotty et al., 2013*). On a large (μm) length scale, branched actin networks drive the plasma membrane forward during cell migration, such that on the scale of individual actin branches, the membrane shape can be thought of as more or less constant (*Keren et al., 2008*; *Mueller et al., 2017*; *Schaus et al., 2007*). However, on a smaller (sub-micron) length scale, branched actin networks deform many cellular membranes as part of organelle and vesicle biogenesis and function (*Rottner et al., 2017*). The relationship between cellular membrane curvature and local actin assembly for each of these 'local' membrane-deformation processes remains relatively unexplored (*Daste et al., 2017*).

Clathrin-mediated endocytosis (CME) is an especially attractive process for studies of actin's role in membrane shape changes due to the relatively complete parts list and available quantitative information about the positions, recruitment timing and biochemical function of many of the participating proteins (*Arasada et al., 2018*; *Idrissi et al., 2012*; *Kaksonen et al., 2005*; *Kaksonen et al., 2003*; *Mund et al., 2018*; *Picco et al., 2015*; *Sochacki et al., 2017*; *Taylor et al., 2011*). CME is a ubiquitous and essential cellular process by which cells take macromolecules from the extracellular space and the plasma membrane into the cell interior (*Kaksonen and Roux, 2018*). During CME, the

**eLife digest** The outer membrane of a cell is a tight but elastic barrier that controls what enters or leaves the cell. Large molecules typically cannot cross this membrane unaided. Instead, to enter the cell, they must be packaged into a pocket of the membrane that is then pulled inside. This process, called endocytosis, shuttles material into a cell hundreds of times a minute.

Endocytosis relies on molecular machines that assemble and disassemble at the membrane as required. One component, a protein called actin, self-assembles near the membrane into long filaments with many repeated subunits. These filaments grow against the membrane, pulling it inwards. But it was not clear how actin filaments organize in such a way that allows them to pull on the membrane with enough force – and without a template to follow.

Akamatsu et al. set about identifying how actin operates during endocytosis by using computer simulations that were informed by measurements made in living cells. The simulations included information about the location of actin and other essential molecules, along with the details of how these molecules work individually and together. Akamatsu et al. also developed a method to count the numbers of molecules of a key protein at individual sites of endocytosis. High-resolution imaging was then used to create 3D pictures of actin and endocytosis in action in human cells grown in the laboratory.

The analysis showed the way actin filaments arrange themselves depends on the starting positions of a few key molecules that connect to actin. Imaging confirmed that, like a pole-vaulting pole, the flexible actin filaments bend to store energy and then release it to pull the membrane inwards during endocytosis. Finally, the simulations predicted that the collection of filaments adapts its shape and size in response to the resistance of the elastic membrane. This makes the system opportunistic and adaptable to the unpredictable environment within cells.

plasma membrane is bent, pinched, and pulled inward in a time frame of ~60 s thereby transitioning from a flat sheet into a spherical vesicle ~100 nm in diameter. Clathrin and its adaptor proteins establish a coat that generates initial membrane curvature (*Chen et al., 1998*; *Pearse, 1976*; *Stachowiak et al., 2012*), and BAR (bin-amphiphysin-rvs)-domain proteins bind curved membranes and support further membrane curvature (*Buser and Drubin, 2013*; *David et al., 1996*; *Kishimoto et al., 2011*). During yeast endocytosis, branched actin filaments provide the force required for membrane tubule formation (*Engqvist-Goldstein and Drubin, 2003*; *Idrissi et al., 2012*; *Kukulski et al., 2012*; *Picco et al., 2018*; *Sun et al., 2006*; *Wang and Carlsson, 2017*). In metazoan cells, endocytic pits under high tension stall at a 'U'-shaped intermediate in the absence of functional actin (*Boulant et al., 2011*), implying that actin is required to generate plasma membrane shape changes late in CME (*Hassinger et al., 2017*; *Yarar et al., 2005*; *Yoshida et al., 2018*). The molecular mechanism by which a network of polarized, branched actin filaments assembles at these sites for productive force generation is poorly understood.

Actin network assembly is known to play a key role in membrane shape change in some contexts. For example, mathematical modeling (*Berro et al., 2010*; *Carlsson and Bayly, 2014*; *Dmitrieff and Nédélec, 2015*; *Liu et al., 2009*; *Mund et al., 2018*; *Wang et al., 2016*) and quantitative fluorescence imaging in yeast (*Wu and Pollard, 2005*; *Sirotkin et al., 2010*; *Berro and Pollard, 2014*; *Picco et al., 2015*) have established the relationship between actin filament assembly and plasma membrane shape particular to fungi, which have unique mechanical requirements due to very high (~10 atm) hydrostatic turgor pressure. However, less is known about actin organization and function in the lower force regime characteristic of metazoan cells. A multiscale modeling effort that accounts for the mechanics of single actin filaments and that is constrained by experimental measurements of actin dynamics, spatial organization of the filaments, and tension in the plasma membrane is required to gain insight into actin organization and force generation capacity. We hypothesize that in localized membrane-reshaping processes such as endocytosis, branched actin networks assemble under specific spatial 'boundary conditions,' which serve as geometrical constraints dictated both by the shape of the membrane and the spatial segregation of membrane-associated proteins that interact with actin. These unique spatial boundary conditions on a curved surface, combined with the knowledge of numbers of molecules in cells and known reaction rate constants, provide the

necessary information for multiscale modeling and a mechanistic framework to understand the relationship between plasma membrane mechanics and branched actin assembly and mechanics associated with CME.

Using this framework, we sought to answer the following questions: How do branched actin networks assemble, organize, and produce force around an endocytic pit? How does the spatial segregation of Arp2/3 complex activators (*Almeida-Souza et al., 2018*; *Mund et al., 2018*) and actin-binding proteins associated with endocytic coats (*Clarke and Royle, 2018*; *Engqvist-Goldstein et al., 2001*; *Sochacki et al., 2017*) influence this organization? And finally, how do endocytic actin networks adapt to changing loads due to the stochastic environment and changes in membrane tension? To answer these questions, we combined live-cell molecule counting methods in genome-edited diploid human cells and cryo-electron tomography of intact cells with multiscale modeling of plasma membrane mechanics and actin filament dynamics. Our results show that a minimal branched actin network is sufficient to create sustained internalization of an endocytic pit against physiological membrane tension. Actin filament self-organization and bending, which arise from the spatial distribution of actin-coat attachments around the curved endocytic pit, allow the actin network to adapt to changing loads. We anticipate that the mechanistic insights gained for actin in mammalian endocytosis will also apply to a variety of local membrane-bending processes carried out by branched actin throughout the cell.

## Results

### Multiscale modeling shows that a minimal branched actin network is sufficient to internalize endocytic pits against physiological membrane tension

We combined a continuum-mechanics model of the plasma membrane, an agent-based model of actin filament dynamics, quantitative fluorescence microscopy, and electron tomography in cells to determine the molecular mechanism by which branched actin networks produce force during mammalian clathrin-mediated endocytosis (*Figure 1*, *Scheme 1*).

First, we used a continuum-mechanics model of the plasma membrane (*Alimohamadi et al., 2018*; *Hassinger et al., 2017*; *Rangamani et al., 2014*) to determine the force-extension relationship for clathrin-coated pits stalled at a U-shaped intermediate under high membrane tension (*Figure 1B*). Here, the extension refers to the extent of pit internalization, which is a displacement in the -Z direction (*Figure 1A–B*). Previously, we showed that membrane curvature generation by the endocytic coat during vesicle formation could snap the membrane into a pinched 'omega' shape as a function of membrane tension and the curvature induced by the coat (*Hassinger et al., 2017*), but we did not focus on force produced by the actin cytoskeleton. Here, we modeled the coated membrane based on the *Helfrich (1973)* energy and applied a linear force to the clathrin-coated pit in increasing value over successive simulations, corresponding to a simplified actin force. Simulations demonstrated that a clathrin-coated pit experiences a nearly linear force-extension relationship until an internalization of ~100 nm, at which point the pit can also adopt a pinched (or 'omega') shape, which requires a lower force (*Figure 1C* and *Figure 1—video 1*). We calculated the resistance to internalization as the slope of the force-extension plot for the linear regime and found that it is directly proportional to plasma membrane tension for a wide range of coat rigidities (*Figure 1D*). Importantly, this direct scaling between resistance to internalization and membrane tension allowed us to treat this step of endocytic pit internalization as a linear spring, with the spring constant calibrated using measurements of plasma membrane tension in mammalian cells (*Diz-Muñoz et al., 2016*; Kaplan et al., in preparation).

The simple spring-like relationship uncovered above between force and endocytic pit internalization (*Figure 1D*) allowed us to simplify our mechanical treatment of the plasma membrane while modeling individual actin filaments and actin-binding proteins with realistic kinetics and mechanics (*Figure 1E–G*, *Supplementary file 3*). We used Cytosim (*Nedelec and Foethke, 2007*) to construct a filament-based model of the endocytic actin network. This agent-based model allowed us to simulate the emergent architecture and mechanical functions of branched actin for realistic endocytic geometries.

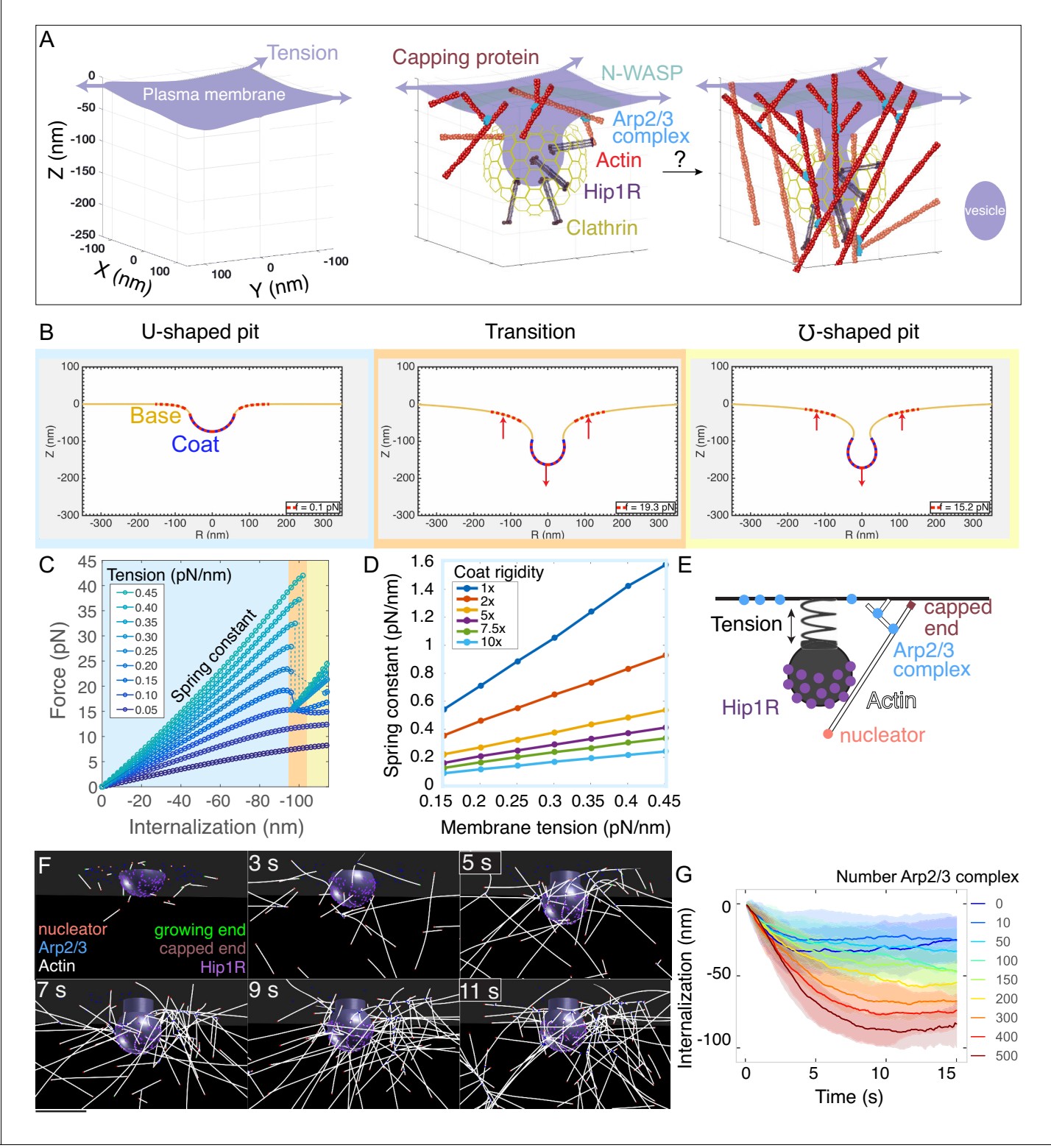

**Figure 1.** Multiscale modeling shows that a minimal branched actin network is sufficient to internalize endocytic pits against physiological membrane tension. (**A**) Schematic of a section of the cell's plasma membrane being internalized during mammalian endocytosis depicts plasma membrane deformation against membrane tension (purple arrows) countered by the clathrin coat (yellow) and the actin cytoskeleton (red). (**B**) Shape of the membrane and pit internalization from continuum mechanics simulations of the endocytic pit experiencing axial (Z) forces corresponding to simplified actin forces. To begin with, the plasma membrane (yellow) is deformed by a coat with preferred curvature that expands in area until the pit stalls. A net

*Figure 1 continued on next page*

*Figure 1 continued*

force (red arrows) is applied downward from the coat and upward into the base of the endocytic pit (red dotted lines). In this simulation, membrane tension was 0.2 pN/nm, and the coated area was rigid (2400 pN·nm). (C) Force versus pit internalization relationships for different values of membrane tension. Internalization is defined as the pit displacement in Z. Shading delineates linear force-internalization regime (blue); 'transition point' from U to omega shape (orange); 'omega-shaped' regime where the neck is narrower than the pit diameter and the force required for internalization is lower than at the transition point (for tensions > 0.1 pN/nm) (yellow). Color matches the three snapshots in B. Parameters are given in **Supplementary files 1** and **2**. (D) Resistance of pit to internalization versus membrane tension. Resistance (spring constant) is defined as absolute value of slope in C for the 'U-shaped' region. Each curve is calculated for a different value of membrane rigidity (where 1x = 320 pN·nm, the rigidity of the uncoated plasma membrane). (E) Computational model of branched actin filament polymerization coupled to endocytic pit internalization. An internalizing endocytic pit is modeled as a sphere with a neck attached to a flat surface by a spring. Active Arp2/3 complex (blue) is distributed in a ring around the base of the pit. An actin nucleation protein (pink) generates an actin filament (white), which polymerizes, stalls under load, and is stochastically capped (red). Arp2/3 complexes bind to the sides of actin filaments and nucleate new filaments at a 77-degree angle, creating new branches. Linker Hip1R (purple) is embedded in the pit and binds to actin filaments. Model parameters are given in **Supplementary file 3**. (F) Graphical output of the simulations from Cytosim (**Nedelec and Foethke, 2007**) at 2 s intervals. Scale bar: 100 nm. (G) Pit internalization over simulated time as a function of the number of available molecules of Arp2/3 complex. Average of 16 simulations per condition. Shaded bars are standard deviations.

The online version of this article includes the following video and figure supplement(s) for figure 1:

**Figure supplement 1.** Effect of different actin- and simulation-related parameters on pit internalization dynamics.
**Figure supplement 2.** Initiation from a pool of diffusing cytoplasmic actin filaments leads to variable timing of internalization.
**Figure 1—video 1.** Simulations of continuum membrane mechanics model.
https://elifesciences.org/articles/49840#fig1video1
**Figure 1—video 2.** Simulation of actin in endocytosis using Cytosim.
https://elifesciences.org/articles/49840#fig1video2

---

We simplified the endocytic pit as a solid, impermeable structure, initially a hemisphere, attached to a flat plasma membrane corresponding to the 'U-shaped' intermediate (**Avinoam et al., 2015**; **Boulant et al., 2011**; **Messa et al., 2014**; **Yarar et al., 2005**; **Figure 1E**). The following rules were prescribed for actin filament dynamics. Initially, actin filament-nucleating proteins seed a small number of actin filaments near the endocytic pit. These randomly-oriented 'mother filaments' serve as templates for binding pre-activated Arp2/3 complexes, which correspond to the coincidence of Arp2/3 complex and its activator N-WASP, arranged in a ring (**Almeida-Souza et al., 2018**; **Mund et al., 2018**) at the base of the endocytic pit (**Idrissi et al., 2008**; **Kaksonen et al., 2003**; **Picco et al., 2015**; Kaplan et al., in preparation). When an active Arp2/3 complex comes in proximity

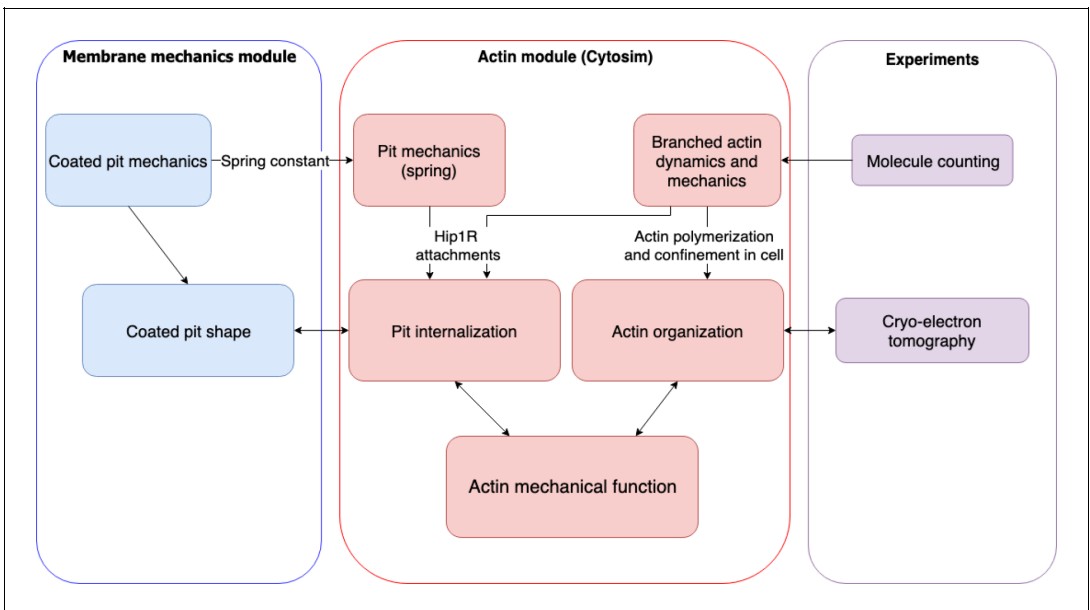

**Scheme 1.** Flow chart of multiscale modeling and experimental strategy combining membrane mechanics, actin spatiotemporal dynamics, molecule counting, and cryo-electron tomography.

with an actin filament, it can bind to the filament and nucleate the growth of a new branched filament at an ~77° angle (*Blanchoin et al., 2000*). Growing actin filaments can polymerize, diffuse under thermal fluctuations, and bend under force, and their growing ends are capped stochastically. Filament growth decreases with load according to the Brownian ratchet mechanism (*Mogilner and Oster, 1996*; *Peskin et al., 1993*). Growth of the actin network is coupled to internalization of the endocytic pit by an actin-linking protein (Hip1/Hip1R/Epsin, simplified here as Hip1R), which is embedded in the coated pit and binds to actin filaments (*Clarke and Royle, 2018*; *Engqvist-Goldstein et al., 2001*; *Engqvist-Goldstein et al., 1999*; *Sochacki et al., 2017*). Importantly, most of the parameters in this model have been determined with measurements in vitro or in vivo, including the dimensions of the endocytic pit, its resistance to internalization (modeled as a spring, *Figure 1D*), rates of association and dissociation of different proteins, branching angles, capping rates, filament persistence length, and stall force (*Supplementary file 3* and Materials and methods).

Stochastic simulations of the model showed that this minimal branched actin network internalizes endocytic pits up to ~60 nm against physiological membrane tension (*Figure 1F* and *Figure 1—video 2*). In order to compare different conditions, we used two metrics – internalization of the pit (in nm) over time (*Figure 1G*) and the 95th percentile of internalization (*Figure 1—figure supplement 1A*). Then, we evaluated the robustness of the model to different parameters by conducting a series of parameter sweeps (*Figure 1—figure supplement 1*). We found that the extent of internalization is robust to a wide range of parameters, including filament stiffness, stall force, and affinity between Hip1R attachments and actin filaments (*Figure 1—figure supplement 1*). Initiating the simulations from a cytoplasmic pool of linear actin filaments (*Raz-Ben Aroush et al., 2017*) allowed for endocytosis but the timing of the onset of internalization was more variable (*Figure 1—figure supplement 2*). The extent of internalization was particularly sensitive to the number of available Arp2/3 complexes (*Figure 1G*), indicating a need for precise measurements of this molecule at mammalian endocytic sites.

## Molecule counting of endogenously GFP-tagged Arp2/3 complex in live mammalian cells

Motivated by our prediction that internalization rate is sensitive to the number of Arp2/3 complexes, we developed a method to count the number of molecules of endogenously GFP-tagged proteins in living mammalian cells (*Figure 2*). We adapted the self-assembling GFP-tagged protein nanocages developed by *Hsia et al. (2016)* for expression in live cells to create a fluorescence-based calibration curve relating fluorescence intensity of endogenously GFP-tagged proteins to numbers of molecules per endocytic site. Given that the nanocages are derived from bacterial glycolytic enzymes, we made point mutations known to abolish enzymatic activity of the proteins. To slow the diffusion of the intracellular nanocages and facilitate fluorescence measurements, we introduced an inducible dimerization motif to the plasma membrane by fusing the construct to FKBP and coexpressing a palmitoylated and myristoylated FRB variant (*Figure 2A*). The resulting two-component fusion protein transiently associated with the plasma membrane even without rapamycin analog AP21967, but the extent of association with the plasma membrane increased in a dose-dependent manner with the concentration of AP21967 (*Figure 2—figure supplement 1A–B* and *Figure 3—video 1*). We measured the fluorescence intensity of four GFP-tagged nanocages of copy number ranging from 12 to 120 copies of GFP per structure using spinning disk confocal microscopy (*Figure 2B*). After correcting for exposure time (*Figure 2—figure supplement 1E–F*), uneven illumination intensity, and local background (Materials and methods), the fluorescence intensity per spot was unitary (*Figure 2C*) and directly proportional to the predicted numbers of molecules per structure ($R^2$ = 0.996) (*Figure 2D*). Using this calibration curve, we measured the numbers of molecules of an *E. coli* flagellar motor protein eGFP-MotB, which resulted in measurements similar to previously published measurements (*Figure 2—figure supplement 1G–I*). Thus, we established the suitability of this method to relate fluorescence intensity of endogenously GFP-tagged proteins to numbers of molecules inside live mammalian cells.

To measure the timing, frequency, and numbers of Arp2/3 complexes assembling at sites of clathrin-mediated endocytosis, we used CRISPR/Cas9-mediated genome editing to endogenously tag the C terminus of ArpC3, a subunit of the Arp2/3 complex, with the fluorescent protein tagGFP2 in human induced pluripotent stem cells (*Figure 2E*, *Figure 2—figure supplement 2A–B*). Human

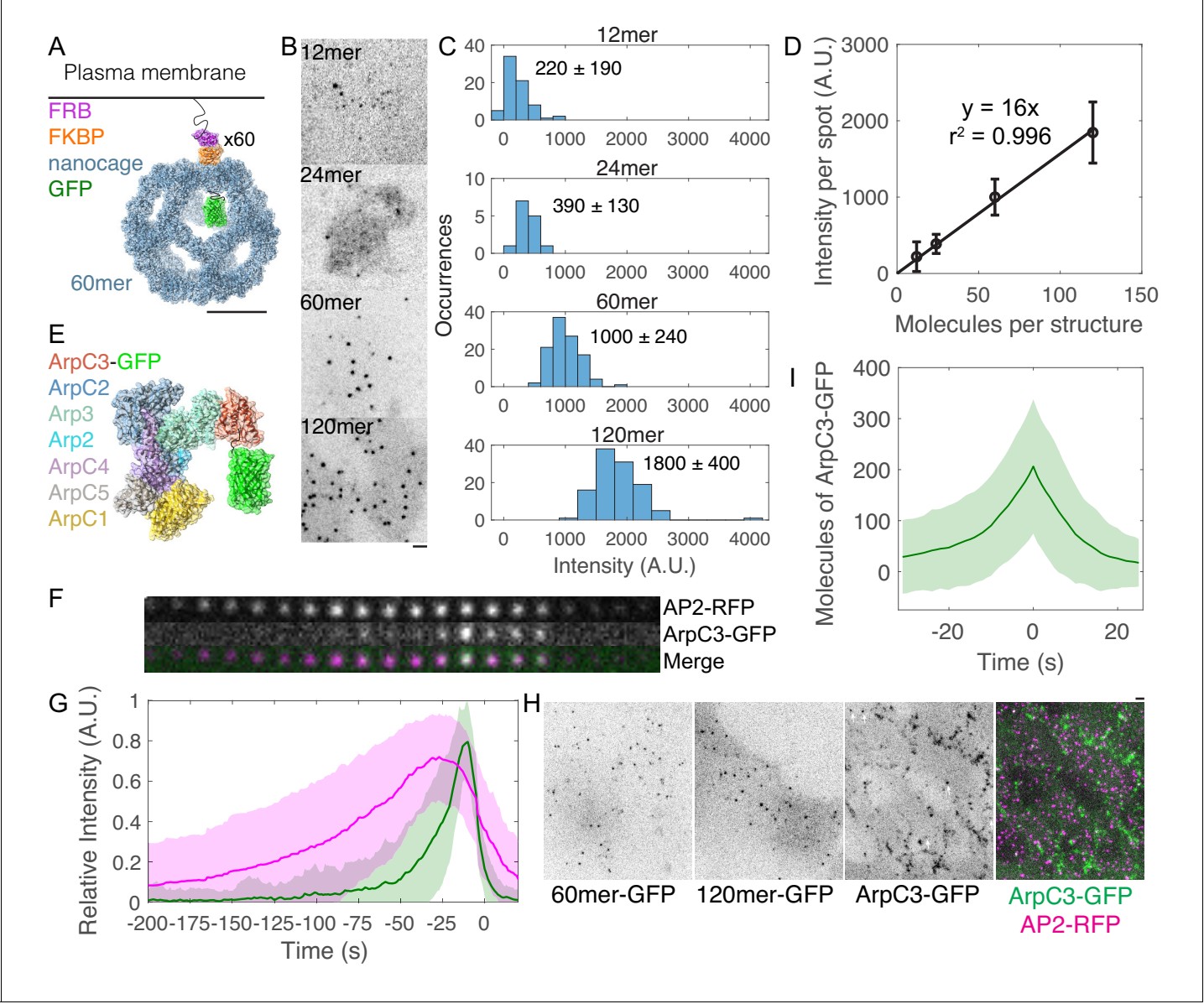

**Figure 2.** Molecule counting of endogenously GFP-tagged Arp2/3 complex in live human induced pluripotent stem cells. (A–D) Development of a calibration curve relating fluorescence intensity to numbers of molecules in live cells. (A) Cartoon of intracellular GFP-tagged 60mer nanocage with inducible plasma membrane tether. Each subunit (blue) is tagged with GFP (green) and FKBP (orange). FRB (T2098L) (Purple) is targeted to the plasma membrane by a palmitoylation and myristoylation sequence and dimerizes with FKBP in the presence of rapamycin analog AP21967. Cartoon showing one of 60 tagged subunits is based on PDB structures 5kp9, 2y0g, and 4dri. Scale bar 10 nm. (B) Inverse contrast fluorescence intensity images of human induced pluripotent stem cells expressing GFP-tagged plasma membrane-bound nanocages. Sum projection of nine 300 nm confocal images. Scale bar: 2 µm. (C) Histograms of fluorescence intensity per spot for the four calibration constructs showing mean ± standard deviation. Images were corrected for uneven illumination and intensity was background-corrected. Data from 305 spots in 15 cells over three experiments. (D) Calibration curve relating fluorescence intensity to numbers of molecules in mammalian cells. Line is a linear fit through zero. Error bars are standard deviations. (E) Cartoon drawn to scale of Arp2/3 complex tagged with GFP at the flexible C-terminus of ArpC3. Known binding and activation sites are distal to this site. Based on PDB 2p9l. (F) Montage of CME event marked by AP2-tagRFP-T and ArpC3-tagGFP2 from TIRF imaging. Montage shows 4 s intervals from a movie taken at 2 s intervals. (G) Relative fluorescence intensity over time of AP2-tagRFP-T and ArpC3-tagGFP2 in endocytic events imaged by TIRF microscopy. Traces were normalized to maximum intensity and averaged. 121 traces from 8 cells in four experiments. Shading is ±1 s.d. (H) Fluorescence micrographs of (left) 60mer-tagGFP2, (left-center) 120mer-tagGFP2, (right-center) ArpC3-tagGFP2, and (right) ArpC3-tagGFP2 and AP2-tagRFP-T. White arrows mark spots in which ArpC3-tagGFP2 and AP2-tagRFP-T colocalize. Scale bar 2 µm. (I) Numbers of molecules of ArpC3 over time.

The online version of this article includes the following video and figure supplement(s) for figure 2:

*Figure 2 continued on next page*

*Figure 2 continued*

**Figure supplement 1.** Optimization and validation of fluorescence calibration method.
**Figure supplement 2.** Generation of genome-edited human induced pluripotent stem cell lines endogenously expressing AP2-RFP and ArpC3-GFP.
**Figure 2—video 1.** Time lapse images of human induced pluripotent stem cells transiently expressing FKBP-60mer-GFP and treated with 0.5 nM AP21967.
https://elifesciences.org/articles/49840#fig2video1
**Figure 2—video 2.** Time-lapse TIRF microscopy image of a human induced pluripotent stem cell endogenously expressing ArpC3-GFP and AP2-RFP.
https://elifesciences.org/articles/49840#fig2video2

induced pluripotent stem cells are diploid and thus suitable for molecule-counting measurements when both alleles of the ArpC3 gene are fused to the gene for GFP. C-terminal GFP tags on ArpC3 are more functional than on other subunits of the Arp2/3 complex (*Egile et al., 2005*; *Picco et al., 2015*; *Sirotkin et al., 2010*; *Smith et al., 2013*). Cells tagged at both alleles of the ArpC3 gene had twice the fluorescence intensity of cells with a single allele tagged, suggesting direct proportionality between GFP fluorescence intensity and numbers of molecules (*Figure 2—figure supplement 2C–D*). These cells also endogenously express a tagRFP-T fusion with the µ2 subunit of the adaptor protein AP2, allowing us to identify sites of clathrin-mediated endocytosis (*Hong et al., 2015*).

We determined the relative timing of AP2 and ArpC3 appearance at endocytic sites using time-lapse TIRF imaging and automated two-color particle tracking (*Dambournet et al., 2018*; *Hong et al., 2015*; *Figure 2F*). The vast majority (81 ± 10%, n = 136) of CME events marked by AP2-RFP culminated in a burst of ArpC3-GFP fluorescence, prior to internalization of the pit, persisting until the pit internalized (*Figure 2G* and *Figure 2—video 2*). In addition, 24 ± 4% of ArpC3-GFP tracks (n = 145) did not colocalize with AP2. We hypothesize that these are sites of clathrin-independent endocytosis. Then, using spinning-disk confocal fluorescence microscopy, we compared the fluorescence intensities of ArpC3-GFP spots and GFP-tagged nanocage proteins in cells to determine the numbers of ArpC3-GFP molecules at clathrin-mediated endocytosis sites (*Figure 2H*). Thus, we determined that ~200 molecules of Arp2/3 complex accumulate at clathrin-mediated endocytosis sites over time (*Figure 2I*).

## Self-organization of actin filaments into a radial dendritic network drives endocytic internalization

Incorporating the Arp2/3 molecule number we determined experimentally into our multiscale model, we next conducted simulations of the model to investigate the spatial organization of actin and force generation capacity of the endocytic network (*Figure 3*). Strikingly, the actin network self-organized around the endocytic pit. This self-organized network drove the assembly of 150 ± 30 actin filaments (*Figure 3—figure supplement 1A*) containing 5700 ± 1100 monomers (*Figure 3—figure supplement 1B*). Interestingly, only a small number of actin filaments (<5) grew at any given time because the filaments became capped soon after they were nucleated (*Figure 3—figure supplement 1C*; *Berro et al., 2010*; *Rangamani et al., 2011*; *Xiong et al., 2010*). Filament lengths were exponentially distributed with a final length of 90 ± 80 nm (*Figure 3—figure supplement 1D–E*). Actin filaments bound to 120 ± 10 Hip1R molecules in the coat (*Figure 3—figure supplement 1F*). The endocytic pit internalized ~60 nm in 10–15 s (*Figure 3A and D*). Based on the initial geometry of the endocytic pit and activated Arp2/3 complex, branched actin filaments self-organized into a radial dendritic network: the network attached to the clathrin coat by binding to Hip1R, the pointed (minus) ends localized close to the pit and the barbed (plus) ends near the base of the pit were oriented to grow toward the base of the pit (*Figure 3A–C* and *Figure 3—video 1*).

The axial self-organization of this branched actin network resembles that at the leading edge of cells (*Figure 3—figure supplement 1G–I*; *Maly and Borisy, 2001*; *Mueller et al., 2017*; *Schaus et al., 2007*; *Svitkina and Borisy, 1999*), with an important difference. Because actin filament attachment sites are located on the coat of the endocytic pit, filaments radiate away from the center of the pit, such that most of the barbed ends orient radially away from the center of the pit rather than toward the coat or neck (*Figure 3E*). The radial orientation of barbed ends gradually increases from the center of the pit, where there is no preferred orientation, to the periphery, where the barbed end radial orientation is highest (*Figure 3F*). The extent of radial distribution of the

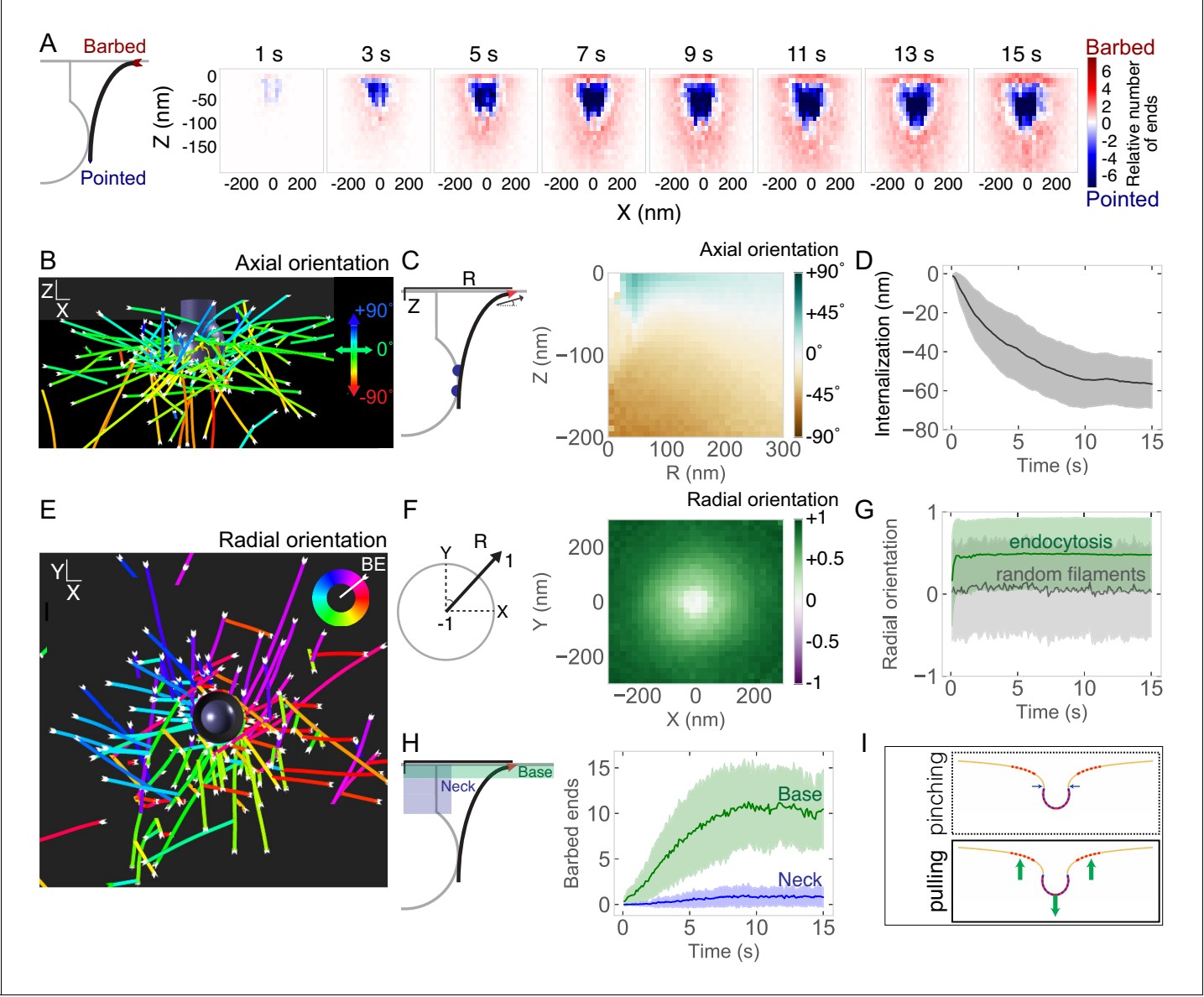

**Figure 3.** Self-organization of actin filaments into a radial dendritic network drives endocytic internalization. (A) (Left) Schematic depicting actin barbed (plus) or pointed (minus) ends. (Right) Heat maps of the positions of barbed ends (red) or pointed ends (blue) relative to the endocytic pit. Color code represents the relative number of ends. Each graph is averaged across 96 simulations and 1 s of simulation time. (B) Simulation output of endocytic actin filaments color-coded for axial (Z) orientation. Blue filaments orient toward the base of the pit (+90°) and green filaments orient parallel to the base of the pit (0°). (C) Axial orientation of barbed ends. (Left) Schematic of axes. R is radial position of barbed end. (Right) Heat map of axial orientation of barbed ends as a function of R and Z position. Average of 96 simulations. (D) Pit internalization over time (n = 96 simulations). (E) Simulation output of endocytic actin filaments color-coded for radial orientation. (F) Radially oriented endocytic actin filaments. (Left) Schematic of axes. Radial orientation is defined such that +1 = barbed end oriented away from the center of the pit, and −1 = barbed end oriented toward the center of the pit. (Right) Heat map of radial orientation of barbed ends as a function of X and Y position (n = 96 simulations). Barbed ends radiate outward. (G) Radial orientation of barbed ends over time for 96 simulations. Gray curve is negative control of randomly oriented filaments (n = 50 filaments in one simulation). (H) Concentration of barbed ends near the base of the endocytic pit. (Left) Schematic of positions of the neck and base of the pit. (Right) Number of barbed ends near base (green) or neck (blue) of pit, defined as within 7.5 nm of each surface. (I) The majority of forces are directed orthogonal to the base of the pit based on positions of barbed ends in simulations. Shaded bars are standard deviations.

The online version of this article includes the following video and figure supplement(s) for figure 3:

**Figure supplement 1.** Assembly and self-organization of endocytic actin network.

**Figure 3—video 1.** Simulation of actin in endocytosis with actin filaments color coded for axial orientation.

https://elifesciences.org/articles/49840#fig3video1

filaments increases rapidly after time 0 (*Figure 3G*). An important consequence of this self-organization is that, based on the position of Hip1R and the Arp2/3 complex, more barbed filament ends localize near the base (10 ± 4 ends) than near the neck of the endocytic pit (1 ± 1 ends) (*Figure 3H*). These data result in an important prediction from our model: an actin network self-organized as described here will produce an axial force during pit internalization (*Figure 3I*). We predict that the radial dendritic self-organization is a powerful mechanism that makes endocytic actin networks resilient to biochemical and mechanical perturbations.

## Spatial distribution of actin/coat attachments and Arp2/3 complex, but not Arp2/3 complex density, strongly affects actin self-organization and pit internalization

Our finding that self-organized endocytic actin networks grow toward the base of the pit prompted us to explore the molecular mechanism by which actin filaments self-organize. Actin dynamics in association with the endocytic machinery can be thought of as a polymerization engine constrained by two spatial boundary conditions – active Arp2/3 complex at the base of the pit (*Almeida-Souza et al., 2018*; *Idrissi et al., 2008*; *Kaksonen et al., 2003*; *Mund et al., 2018*; *Picco et al., 2015*; Kaplan et al., in preparation) and Hip1R/actin attachments on the curved pit surface (*Clarke and Royle, 2018*; *Engqvist-Goldstein et al., 2001*; *Engqvist-Goldstein et al., 1999*; *Sochacki et al., 2017*; *Figure 4A*). Given that such spatial boundary conditions confer unique mechanical properties and adaptation to loads under flat geometries in vitro (*Bieling et al., 2016*), we aimed to understand how the boundary conditions corresponding to the curved endocytic pit affect endocytic actin organization and internalization. We tested two different scenarios: varying the surface density of Arp2/3 complex at the base of the pit and varying Hip1R surface coverage around the pit itself.

First, we tested whether the surface density of the Arp2/3 complex at the base of the pit affects endocytic internalization because recent studies in vitro and in yeast suggest that the local concentration of Arp2/3 complex activators is critical for the timing of Arp2/3 complex activation and endocytic progression (*Case et al., 2019*; *Sun et al., 2017*). In a series of simulations, we distributed 200 molecules of active Arp2/3 complex in a ring of increasing outer diameter to vary the surface density. Surprisingly, we found that varying the surface density of Arp2/3 complex along the base of the pit by a factor of 20 had little impact on endocytic outcome (*Figure 4—figure supplement 1*). We also explored whether localization of a fraction of Arp2/3 complexes at the neck of the pit provided an additional advantage for the endocytic outcome. In this scenario, we distributed 50 of the 200 molecules of the active Arp2/3 complex near the neck of the pit. We found that localizing some of the active Arp2/3 complex near the neck of the pit did not have an impact on the outcome of simulations (p>0.5) (*Figure 4—figure supplement 2D–E*; *Figure 4—figure supplement 2*).

We next conducted a series of simulations in which we varied the surface distribution of a constant number of Hip1R molecules to cover between 1% (localized to the tip of the pit) and 80% (up to the neck of the pit) of the pit (*Figure 4B*) and found that the surface distribution of Hip1R around the endocytic pit strongly impacted endocytic outcome (*Figure 4*). Simulations in each of these conditions revealed that endocytic internalization depends on the surface distribution of actin-coat attachments around the endocytic site (*Figure 4C* and *Figure 4—video 1*). Both the rate and extent of internalization increased with increasing surface area of Hip1R around the pit (*Figure 4D*). From a functional standpoint, increased Hip1R surface coverage around the pit drove more barbed ends toward the base of the pit (*Figure 4E*). This increase in Hip1R surface coverage resulted in an increase in Arp2/3 complexes bound in the endocytic actin network (*Figure 4F*), which in turn nucleated more actin filaments (*Figure 4G*). Simulations showed that a threshold of ~100 Hip1R molecules on the pit is necessary for endocytic internalization (*Figure 4—figure supplement 3A*). The high impact of Hip1R surface distribution on actin filament organization implies that Hip1R molecules distributed broadly around the pit allow for multivalent attachments between the pit and actin filaments, resulting in filaments being captured in an orientation conducive to force production.

Further examination of the simulations revealed that the Hip1R surface distribution supports a self-organized dendritic actin network via a mechanism of stochastic self-assembly and selection for actin filaments growing toward the base of the pit (*Figure 4—figure supplement 3B*). Mother filaments initially bind and unbind the coat in random orientations (*Figure 4—figure supplement 3B–C*). Filaments growing toward the interior of the cell do not template the growth of new branched

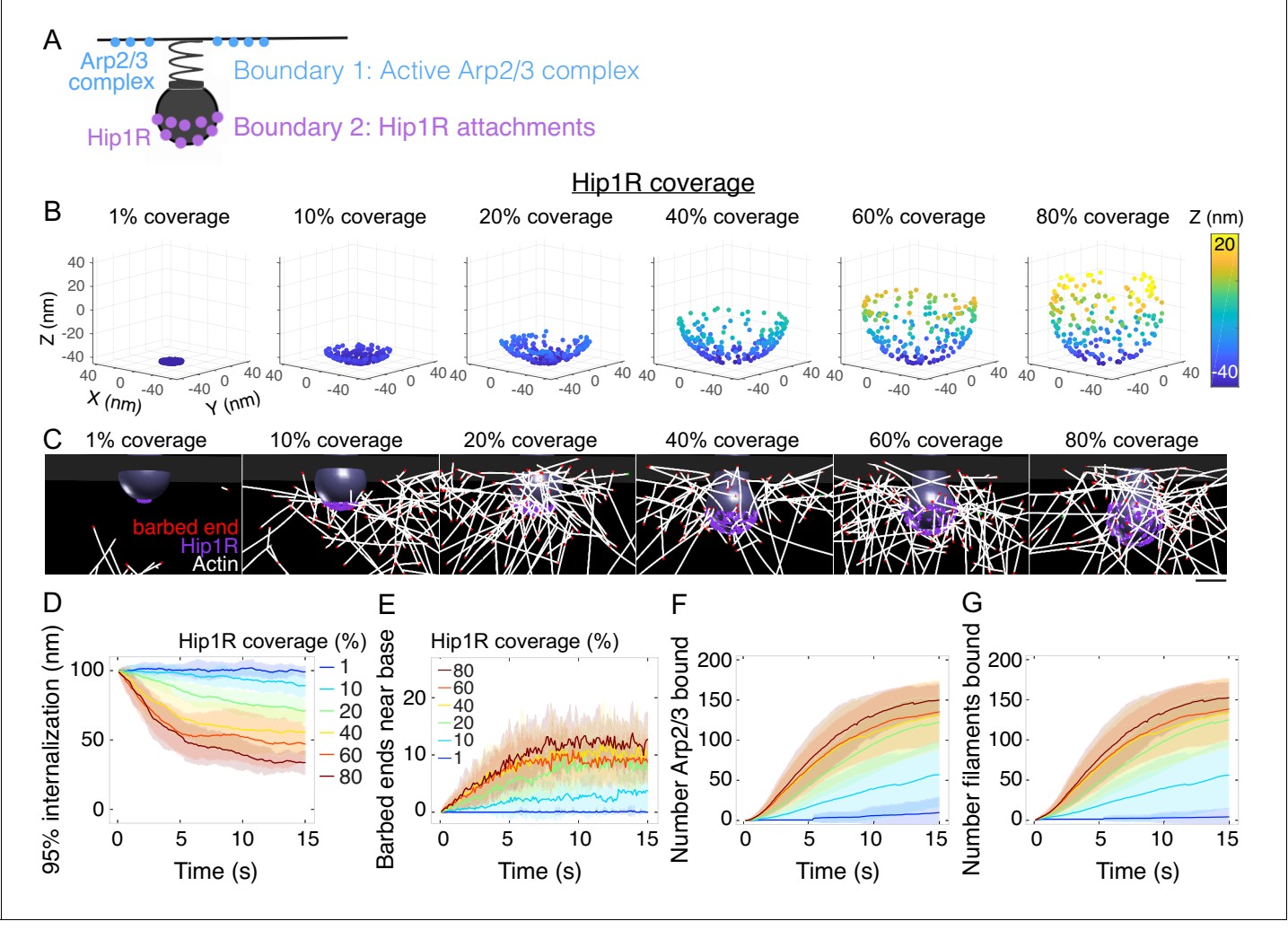

**Figure 4.** Spatial distribution of actin/Hip1R attachments strongly affects actin self-organization and pit internalization. (**A**) Schematic of spatial boundary conditions from endocytic actin-binding proteins. Positions of active Arp2/3 complex (blue) and actin/pit attachments via linker proteins such as Hip1R (purple). (**B**) Initial positions of Hip1R around increasingly large pit surface area, from 1% to 80% of a sphere. The top ~20% of the sphere is occluded by the neck. (**C**) Snapshots of a series of simulations for different values of Hip1R coverage showing actin distribution at t = 13 s. (**D–G**) Changes in the endocytic actin network over time as a function of Hip1R coverage (colors). n = 96 simulations. (**D**) Internalization; (**E**) Number of barbed ends near the base of the pit (within 7.5 nm); (**F**) Number of Arp2/3 complexes bound in the endocytic network; (**G**) Number of actin filaments bound in the endocytic network. Scale bar: 50 nm.

The online version of this article includes the following video and figure supplement(s) for figure 4:

**Figure supplement 1.** Relationship between endocytic outcome and active Arp2/3 complex surface density or mother filament nucleating protein surface density at the base of the pit.

**Figure supplement 2.** A collar of active Arp2/3 complex near the neck the pit does not affect endocytic outcome.

**Figure supplement 3.** Internalization as a function of the number of Hip1R molecules and mechanism of self-organization of endocytic actin filaments.

**Figure 4—video 1.** Simulations in which the coverage of linker Hip1R around the pit was varied from 1% to 80% of a sphere.

https://elifesciences.org/articles/49840#fig4video1

actin filaments. However, filaments growing toward the base of the pit encounter active Arp2/3 complex, which catalyzes dendritic nucleation of new actin filaments growing in a similar direction (*Figure 4—figure supplement 3B and D*; *Carlsson, 2001*). As a result, near the base of the pit, filaments increasingly orient toward the base of the pit over time (*Figure 4—figure supplement 3E–F*). Our observations therefore establish an important principle for actin organization in endocytosis: the positions of active Arp2/3 complexes are critical for organizing the actin network and determining the direction of force production, while the Hip1R linker distribution is critical for recruiting

mother filaments that activate the Arp2/3 complex to direct filament growth to the area surrounding the base of the pit.

## Bending of endocytic actin filaments contributes to endocytic robustness

Given that self-organized actin filaments help to nucleate new actin filaments that grow toward the base of the pit, questions remained about how these filaments contribute to and sustain force production. Closer examination of the simulations revealed that long actin filaments bend between their attachment sites in the clathrin coat and the base of the pit as a result of assembly confined by the membrane at the base of the pit (*Figure 5A*). We predict that these bent filaments provide a previously unrecognized means of force production by endocytic actin filaments. To test the prediction that actin filaments bend at sites of mammalian endocytosis, we used cryo-electron tomography on intact vitrified mammalian (SK-MEL-2) cells. SK-MEL-2 cells grown on electron-microscopy grids are thin at their periphery (<1 μm), which makes them suitable for electron tomography studies. Indeed, we found bent actin filaments present at sites of clathrin-mediated endocytosis, between the clathrin coat (*Figure 5—figure supplement 1*) and the base of the pit (*Figure 5B* and *Figure 5—video 1*), in extended 'U'-shaped clathrin-coated pits similar to the stage modeled in our simulations (*Figure 5C–D*).

What could be a functional consequence of such bent filaments? We hypothesized that the bent actin filaments store elastic energy that promotes endocytic internalization. We first quantified the filament bending in simulations and found that many (13 ± 3%) of the actin filaments bend further than can be accounted for by thermal fluctuations (*Boal and Boal, 2012*; *Mogilner and Oster, 1996*; *Figure 5E* and *Figure 5—figure supplement 2A*). Most (92%) of the bent filaments bent less than the minimum energy expected to sever the filaments (*De La Cruz et al., 2015b*; *Sept and McCammon, 2001*; *Figure 5E*). Importantly, the bent filaments stored elastic energy by collectively continuing to bend over time, storing up to ~750 pN·nm of elastic energy – mostly in capped filaments (*Figure 5F*). In the context of pit internalization, the amount of elastic energy stored was larger than the magnitude of work required to internalize endocytic pits (*Figure 5F* and *Figure 5—figure supplement 2B*). The elastic energy stored in bent filaments was ~1% of the total energy required to polymerize the endocytic actin network (*Figure 5—figure supplement 2C–D*), with pit internalization constituting ~0.5% of the total energy from actin filament assembly (*Figure 5—figure supplement 2E*). The majority (62 ± 20%) of filament bending energy came from filaments directly bound to Hip1R, and 78 ± 25% of the bending energy came from filaments with barbed ends > 5 nm from the coat surface (*Figure 5—figure supplement 2F–H*). 17 ± 16% of bending energy came from filaments with barbed ends near the base of the pit (*Figure 5—figure supplement 2I*). For filaments near the base of the pit, the bending energy was distributed radially such that filaments with barbed ends ~ 130 nm from the center of the pit contribute the most bending energy (*Figure 5—figure supplement 2J*).

Filament bending serves as an important functional consequence of the self-organization of actin filaments at endocytic sites (*Figure 4*). With high Hip1R surface coverage around the pit, filaments directed to grow toward the base of the pit bend, storing elastic energy (*Figure 5—figure supplement 2K–L*). This elastic energy can be harnessed gradually under thermal fluctuations to drive endocytic internalization through a ratchet mechanism (*Mogilner and Oster, 1996*).

To test the hypothesis that energy stored in bent actin filaments can promote endocytic internalization, we conducted simulations in which the resistance from membrane tension was released at a late time point (t = 10 s, internalization ~50 nm) along with capping filament growth (*Figure 5G*). This scenario allowed us to test how the stored energy in the bent filaments (rather than force generated by growing filaments) can promote internalization in response to an abrupt decrease in tension. We found that once membrane tension decreases, pit internalization sharply increases (*Figure 5H* and *Figure 5—video 2*) and filament bending near the base of the pit gradually decreases by 50% with wide variance (*Figure 5I*). Thus, we found that in addition to generating force actively by filament growth (*Figure 3*), the endocytic actin network stores potential energy in the form of bent filaments that can promote internalization even after filaments have stopped growing.

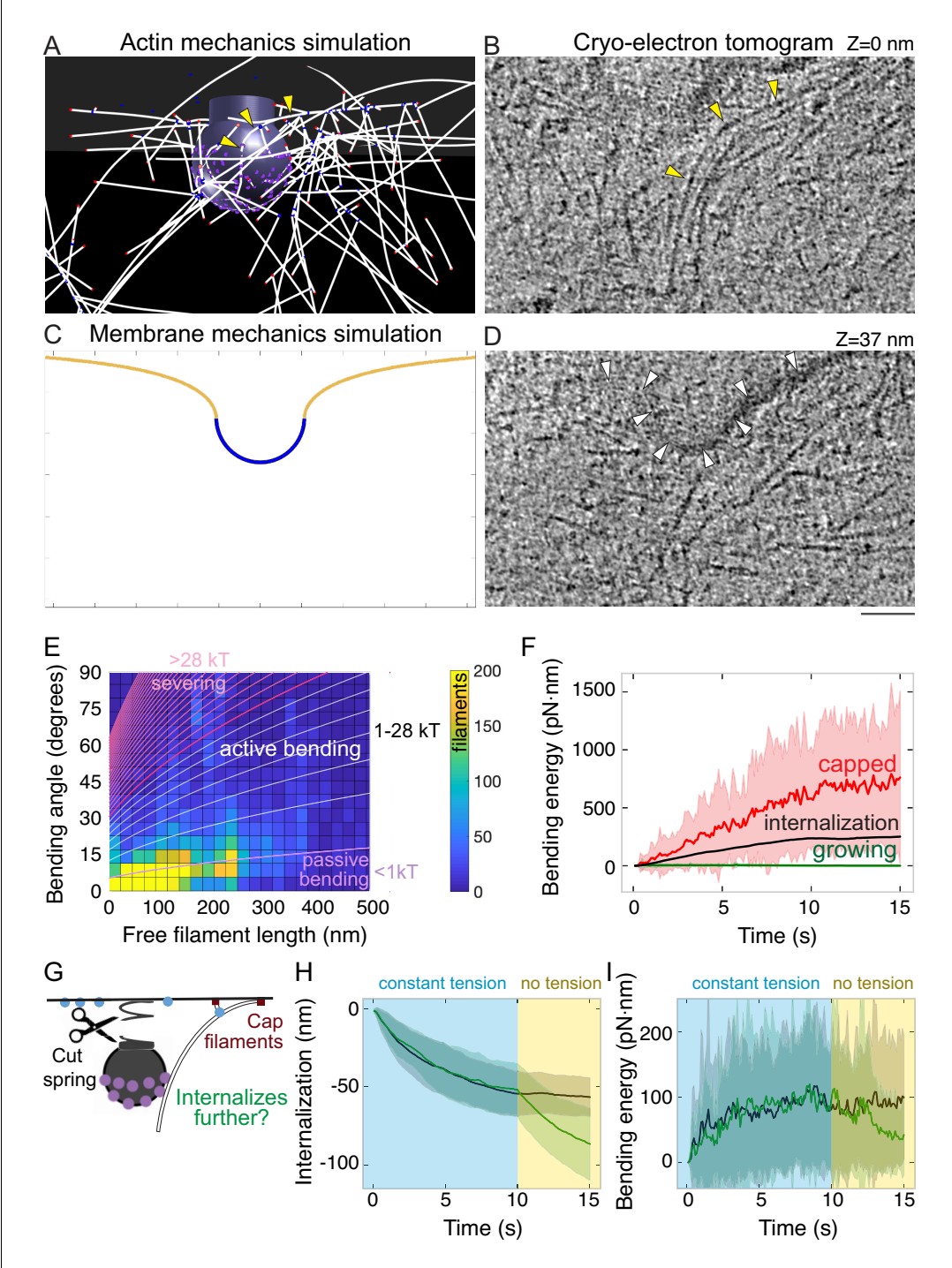

**Figure 5.** Bending of endocytic actin filaments stores elastic energy for pit internalization. (**A**) Snapshot of simulation showing filaments bent between the attachment site in the coat and the base of the pit. Also see *Figure 1F*. Yellow arrowheads point to a bent actin filament. (**B**) Tomographic slice of cryo-electron tomogram of an SK-MEL-2 cell. Long actin filaments (yellow arrowheads) bend along the clathrin-coated pit between the coat and the base of the pit. (**C**) Snapshot of membrane mechanics simulation under an internalization force with 60 nm internalization. (**D**) Slice of the same tomogram as shown in B at a different Z-level (+37 nm) in which the coated plasma membrane (white arrowheads) is visible. Scale bar for A-D: 50 nm. (**E**) Heat map of the bending angle and free filament length of endocytic actin filaments in simulations. Color code is number of filaments (summed for all time points, average of 24 simulations). Lines demarcate the magnitude of energy stored in these filaments, based on the theory of elastic beam rigidity for filaments of persistence length 10 μm (Materials and methods), in units of $k_B T$ (4.1 pN·nm). Purple lines: filament conformations expected from thermal fluctuations (passive bending). White lines: filament bending greater than from thermal fluctuations (active bending). Magenta lines: lower

*Figure 5 continued on next page*

*Figure 5 continued*

limit for bending energy expected to sever filaments (*De La Cruz et al., 2015b*). (F) Total elastic energy stored in bent capped (red) or growing (green) endocytic actin filaments during simulation over time compared to mean energy necessary for internalization (black) (n = 96 simulations). (G) Schematic of an in silico experiment to test the mechanical function of bent endocytic actin filaments. At t = 10 s, the membrane tension was reduced to zero, and the filaments were capped. (H) Internalization (green) after spring cut and filament capping, compared to simulation with no change in tension (black, same data as *Figure 3D*). n = 48 simulations. (I) Bending energy of endocytic actin filaments with barbed ends near base of pit over time. Release of tension and filament capping at t = 10 s (green) compared to no change in tension (black).

The online version of this article includes the following video and figure supplement(s) for figure 5:

**Figure supplement 1.** Hexagonal and pentagonal lattices in tomogram of clathrin-coated pit.

**Figure supplement 2.** Energetics of endocytic actin network.

**Figure 5—video 1.** Cryo-electron tomogram of SK-MEL-2 cell grown on holey carbon grid and vitrified, related to *Figure 5*.

https://elifesciences.org/articles/49840#fig5video1

**Figure 5—video 2.** Simulation of actin in endocytosis in which, at t = 10 s, filaments were all capped and the membrane tension was reduced to 0 pN/nm.

https://elifesciences.org/articles/49840#fig5video2

## Inhibiting Arp2/3 complex activity stalls endocytosis

We next investigated how inhibiting the activity of Arp2/3 complex would affect endocytosis (*Figure 6*). Our simulations, conducted by varying the nucleation rate of Arp2/3 complex, predicted that inhibiting Arp2/3 complex activity stalls endocytosis (*Figure 6A*). Endocytosis was inhibited when Arp2/3 complex nucleation rates fell below the basal value of 1 filament per second (*Beltzner and Pollard, 2008*), and was insensitive to increased rates of nucleation (*Figure 6B*). We validated this relationship with experiments modulating Arp2/3 complex activity in cells. The small molecule inhibitor CK-666 prevents the Arp2/3 complex from nucleating actin filaments (*Hetrick et al., 2013*;

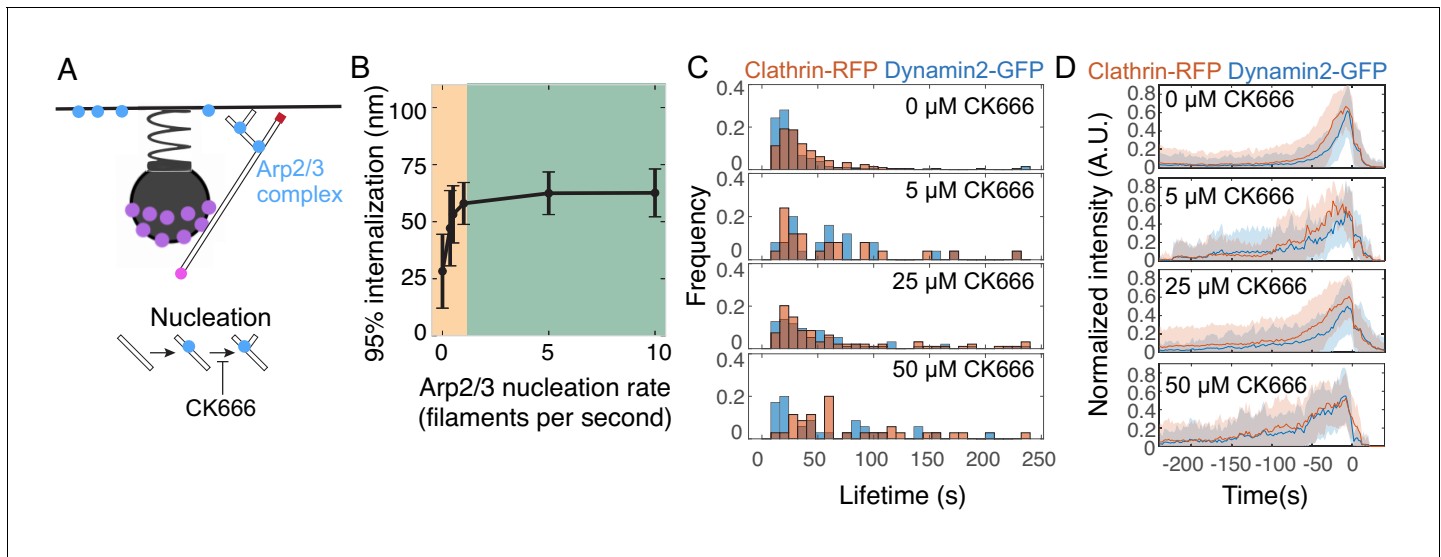

**Figure 6.** Inhibiting Arp2/3 complex nucleation activity stalls endocytosis. (A) Schematic of the model parameter corresponding to Arp2/3 nucleation activity, and the step inhibited by the small molecule CK-666. (B) Internalization as a function of Arp2/3 complex nucleation rate. Orange region highlights parameter sensitivity, and green region highlights parameter insensitivity. n = 96 simulations. Reducing Arp2/3 nucleation rate reduces internalization as seen in the orange region. (C) Histograms of endocytic lifetime in SK-MEL-2 cells endogenously expressing clathrin light chain CLTA-tagRFP-T and dynamin2-eGFP and treated with CK-666. n = 368 tracks from 10 cells. (D) Fluorescence intensity over time for endocytic tracks marked by clathrin-RFP and dynamin2-GFP in SK-MEL-2 cells treated with 0.1% DMSO (0 µM) or the indicated concentration of CK-666 for 45 min. Fluorescence events were tracked automatically (Materials and methods). Tracks in which GFP and RFP colocalized are shown. Each track was normalized to its maximum intensity and then all tracks were averaged and aligned to the time of the disappearance of the clathrin-RFP signal. The lifetimes of these events are plotted in D. Shaded bars are standard deviations.

The online version of this article includes the following figure supplement(s) for figure 6:

**Figure supplement 1.** Effect of Arp2/3 complex inhibitor CK-666 on lifetimes of endogenously tagged markers of endocytosis.

*Nolen et al., 2009*). Treatment of SK-MEL-2 cells with CK-666 inhibited endocytic progression, as marked by the lifetimes of endogenously tagged AP2-RFP or dynamin2-GFP at endocytic sites, in a dose-dependent and time-dependent manner (*Figure 6C–D* and *Figure 6—figure supplement 1A*).

## Adaptation of the endocytic actin network to changes in membrane tension

Because we and others previously modeled that membrane tension plays an important role in membrane bending during the formation of an endocytic pit (*Hassinger et al., 2017*; *Rangamani et al., 2014*; *Walani et al., 2014*) we next varied the value of membrane tension in simulations to understand the relationship between tension, actin filament bending, and actin assembly (*Figure 7A*). In our simulations, endocytic progression attenuated in a tension-dependent manner (*Figure 7A–E*), consistent with previous modeling (*Hassinger et al., 2017*) and experimental observations (*Boulant et al., 2011*; *Ferguson et al., 2017*; *Ferguson et al., 2016*; *Wu et al., 2017*). However, at higher membrane tensions, endocytosis persisted better than expected for a non-adapting network, suggesting the existence of an adaptive mechanism (*Figure 7E*). Therefore, we sought to understand how endocytic actin networks adapt to increases in load.

We found that under low tension (0.015 pN/nm), endocytic pits internalize strongly (*Figure 7B and E*) and few barbed ends encounter the base of the pit (*Figure 7F*), with fewer Arp2/3 complexes recruited to the network (*Figure 7—figure supplement 1A*) and a correspondingly low filament bending energy (*Figure 7H*). Under >50 x higher membrane tension (1 pN/nm), endocytic

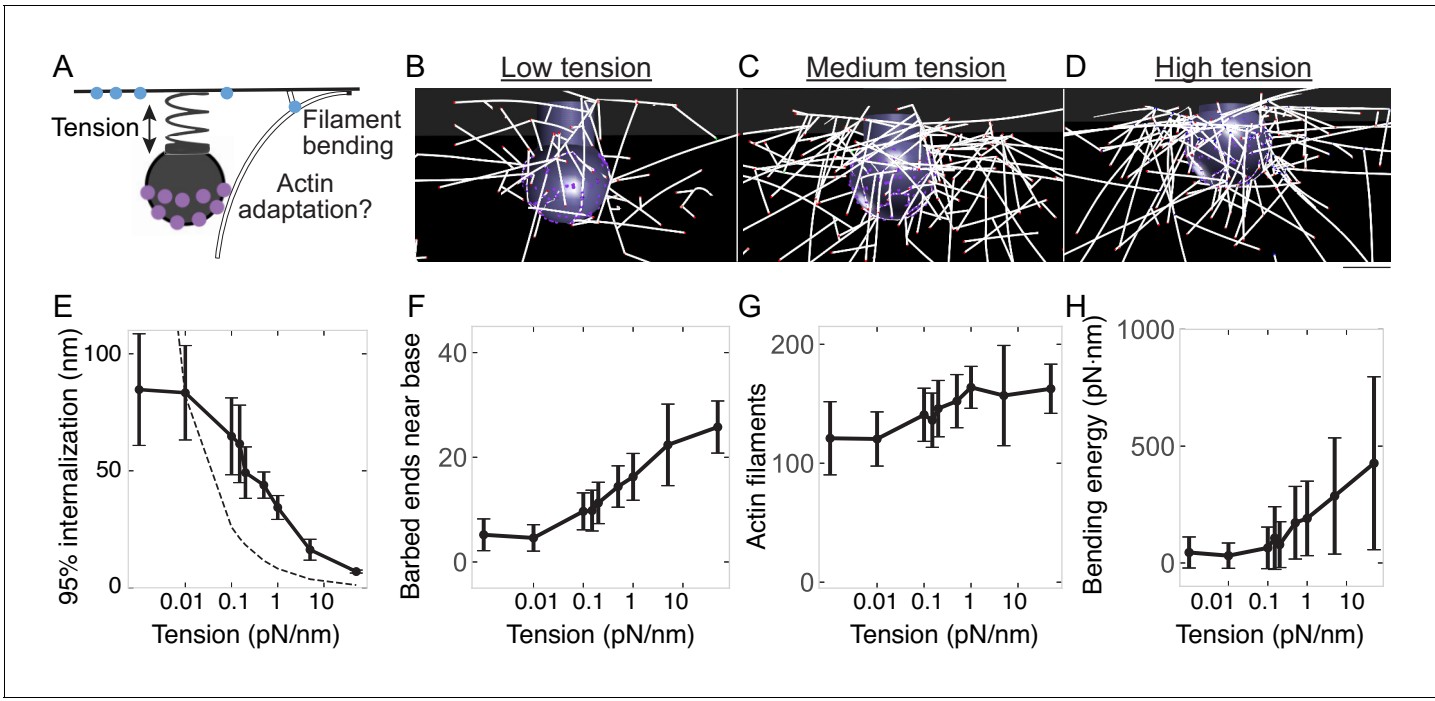

**Figure 7.** Adaptation of endocytic actin network to changes in membrane tension. (**A**) Schematic depicting possible adaptation of the actin network to membrane tension via self-organization and bending. (**B–D**) Snapshots of simulations from the same time point (14 s) for (**B**) low membrane tension (0.015 pN/nm); (**C**) medium membrane tension (0.15 pN/nm); (**D**) high membrane tension (1 pN/nm). Scale bar is 50 nm. (**E–H**) Changes in the endocytic actin network as a function of membrane tension. n = 144 simulations. (**E**) Internalization; (**F**) Number of barbed ends near base of pit; (**G**) Number of actin filaments in Hip1R-bound network; (**H**) Bending energy for filaments with barbed ends near base of pit. Mean ± standard deviation of time points in the last 5 s of simulations. Dashed line in (**E**) is expected internalization based on constant energy usage with 0.01 pN/nm condition as reference (see Methods).

The online version of this article includes the following video and figure supplement(s) for figure 7:

**Figure supplement 1.** Membrane tension-dependent adaptation by the actin network.

**Figure 7—video 1.** Simulations of actin in endocytosis with different values of membrane tension (Low, 0.01 pN/nm; Medium, 0.15 pN/nm; High, 1 pN/nm).

https://elifesciences.org/articles/49840#fig7video1

internalization slowed but was not abolished (*Figure 7E*). For these pits, more barbed ends encountered the base of the pit (*Figure 7F*), binding more Arp2/3 complexes (*Figure 7—figure supplement 1A*) to nucleate more actin filaments (*Figure 7G*) and increasing the total actin filament bending energy near the base of the pit (*Figure 7H*). As a result, upon increasing membrane tension, the overall endocytic energy efficiency increased (*Figure 7—figure supplement 1B*). Thus, the self-organization of the endocytic actin network allows it to adapt to elevated membrane tension by nucleating more filaments at the base of the pit.

### Arp2/3 complex activity and Hip1R/actin attachments are critical for allowing actin filaments to drive endocytic pit internalization and adapt to changing tension

Having established that endocytic internalization depends on two spatially confined boundary conditions – Hip1R/actin attachments at the curved pit (*Figure 4*) and active Arp2/3 complex activity at the base of the pit (*Figure 6*) – we next investigated how these boundary conditions alter the endocytic response to membrane tension (*Figure 8A*).

We systematically varied membrane tension and Arp2/3 complex activity in our model to generate a phase diagram of endocytic internalization as a function of membrane tension and Arp2/3 complex activity (*Figure 8B*). This phase diagram shows that cells with high membrane tension are especially sensitive to changes in Arp2/3 complex nucleation rate (Kaplan et al., in preparation), whereas cells with low membrane tension carry out endocytosis even with low Arp2/3 complex activity, consistent with experimental observations (*Boulant et al., 2011*).

We hypothesized that actin network self-organization arising from the broad Hip1R distribution around the pit (*Figure 4*) and filament bending (*Figure 5*) might allow for the endocytic actin network to change its organization and force-producing capacity under elevated loads (*Figure 7*). To test this hypothesis, we conducted simulations in which Hip1R coverage was varied for different values of plasma membrane tension (*Figure 8A* and *Figure 7—video 1*). We found that the endocytic actin network's ability to adapt to load (*Figure 7*) depends on Hip1R coverage around the pit (*Figure 8D–F*). As the coverage of Hip1R around the pit increased, actin's ability to adapt to changes in membrane tension also increased, as measured by the number of barbed ends near the base of the pit (*Figure 8D*), the binding of active Arp2/3 complex at the base of the pit (*Figure 8E*), subsequent nucleation of additional actin filaments (*Figure 8F*), and bending of actin filaments near the base of the pit (*Figure 8G*). We conclude that sufficient Hip1R coverage around the pit (*Clarke and Royle, 2018*; *Sochacki et al., 2017*) allows endocytic actin filaments to orient in such a way that they can encounter more Arp2/3 complexes at the base of the pit to nucleate more actin filaments. This spatial organization allows the actin network to adapt to sustain force production under a range of opposing loads (*Figure 8H*).

## Discussion

Understanding the relationship between actin filament assembly, actin network organization, and force generation on the plasma membrane requires iterative feedback between experimental measurements and computational modeling. An ultimate goal of this study was to relate single actin filament mechanics to force generation by the collective actin filament network in CME (*Lacayo et al., 2007*). We integrated modeling and quantitative cellular measurements to show that a minimal actin network composed of actin, the Arp2/3 complex and capping protein, with linker attachments in the clathrin coat and rates constrained by cellular and biochemical measurements, is able to generate sufficient force to internalize endocytic pits against mammalian plasma membrane tension. Approximately 200 Arp2/3 complexes constitutively assemble at sites of endocytosis in human induced pluripotent stem cells. Endocytic actin filaments self-organize into a radial dendritic array, in which filaments grow toward the base of the pit. These filaments bend and store elastic energy, which supports internalization. The endocytic actin network adapts to changes in membrane tension by driving more filaments to the base of the pit and increasing filament bending, which supports a higher load and nucleates more actin filaments.

Four lines of experimental evidence support our model (*Figure 8—figure supplement 1*). Two pieces of evidence serve as model validation based on published data and two more are based on experiments conducted in this study. Previous experiments from our lab showed that knocking down

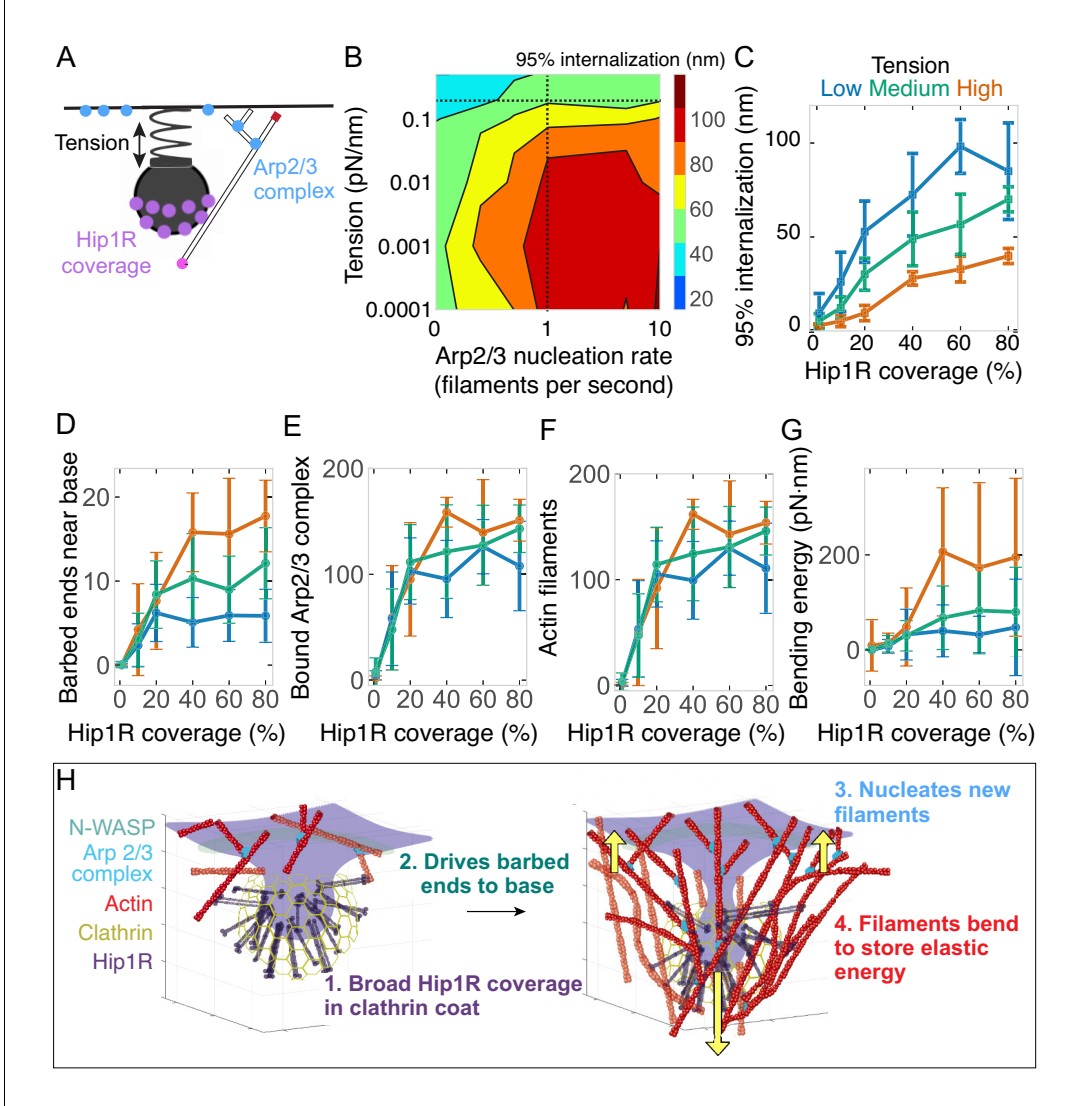

**Figure 8.** Arp2/3 complex activity and Hip1R/actin attachments are critical for allowing actin filaments to drive endocytic pit internalization and adapt to changing tension. (A) Schematic of Arp2/3 complex activity and Hip1R coverage along with membrane tension. (B) Phase diagram of endocytic internalization as a function of membrane tension and Arp2/3 complex nucleation rate shown on a log-log plot. Dotted lines are values taken from the literature (*Beltzner and Pollard, 2008*; *Diz-Muñoz et al., 2016*). (C–G) Changes in the endocytic actin network as a function of Hip1R coverage for different values of membrane tension. Low tension = 0.015 pN/nm; medium tension = 0.15 pN/nm; high tension = 1 pN/nm. n = 288 simulations. (C) Internalization; (D) Number of barbed ends near base of pit; (E) Number of Arp2/3 complexes bound in network; (F) Number of actin filaments bound in network; (G) Bending energy of filaments with barbed ends near the base of the pit. Mean ± standard deviation of time points in the last 5 s of simulations. (H) Summary of load-dependent adaptation of self-organizing endocytic actin network due to spatial segregation of active Arp2/3 complex at the base and Hip1R in a broad distribution within the clathrin coat.

The online version of this article includes the following figure supplement(s) for figure 8:

**Figure supplement 1.** Summary of predictions of the model supported by experimental data in the current manuscript and in the literature.

Hip1R in HeLa cells inhibited endocytosis (*Engqvist-Goldstein et al., 2004*). Consistent with these observations, our simulations showed that a threshold number of actin linkers such as Hip1R is necessary for endocytic internalization (*Figure 4—figure supplement 3A* and *Figure 8—figure supplement 1A–B*). This threshold appears necessary to allow efficient filament capture by the coat and force transmission from the actin network to the coat. Second, experiments showed that capping actin filament elongation with the small molecule compound Cytochalasin inhibits CME, reflected by an increase in stalled endocytic sites marked by clathrin-RFP (*Brady et al., 2010*) or slower

accumulation of dynamin2-GFP at endocytic sites (*Grassart et al., 2014*). We also showed in our model that capping rate is an important parameter for progression of CME; our simulations show that increasing the capping rate of actin filaments inhibits CME, presumably because increasing capping decreases the total amount of actin (*Figure 1—figure supplement 1C* and *Figure 8—figure supplement 1C–D*). In this study, our simulations predicted that actin filaments bend around endocytic pits. These bent filaments store elastic energy for subsequent force production much as a pole vaulter's pole bends and stores energy for delayed force production. Using cryo-electron tomography of intact cells, we observed bent actin filaments at sites of endocytosis in mammalian cells (*Figure 5* and *Figure 8—figure supplement 1E–F*). Finally, we also predicted that inhibiting Arp2/3 complex activity below its basal nucleation rate of 1 filament per second inhibits endocytosis in silico and this prediction was validated in cells using pharmacological agents (*Figure 6C,D* and *Figure 8—figure supplement 1G–H*). Without sufficient Arp2/3 complex, CME fails due to insufficient force production.

Three main conclusions resulted from our study. First, we found that the spatial segregation of Arp2/3 complex activation and Hip1R linker proteins on the clathrin coat are important factors for effective force generation. Unlike actin organization at the leading edge of a migrating cell wherein only one boundary condition at the plasma membrane is sufficient to enable force-generation capacity to be inferred (*Abercrombie, 1980*; *Bieling et al., 2016*; *Mogilner and Edelstein-Keshet, 2002*), in CME two boundary conditions are required – one at the base of the pit for actin polymerization against the plasma membrane and the second on the coat of the pit for attachment of the growing actin filaments so forces are transmitted to the pit to drive internalization. In our model, we used Hip1R as a surrogate for Hip1, Hip1R, and Epsin1/2/3, which cooperatively link actin filaments to the coat (*Brett et al., 2006*; *Chen et al., 1998*; *Messa et al., 2014*; *Senetar et al., 2004*; *Skruzny et al., 2012*). We conclude that based on the relative positions of Arp2/3 complex activators and actin filament linkers, the resultant self-organized actin network orients to produce force orthogonal to the base of the pit rather than producing a pinching force on the neck (*Collins et al., 2011*; *Hassinger et al., 2017*). Pinching forces are achieved by the spontaneous curvature of the rigid coat (*Alimohamadi et al., 2018*; *Foret, 2014*; *Hassinger et al., 2017*). Any constriction forces generated by actin polymerization at the neck would likely occur at a later stage of endocytosis than is the focus of our model, and the filaments would need to be nucleated by a spatially distinct set of Arp2/3 activating proteins around the neck, or by an interaction between other actin filaments and dynamin, but the mechanism for arranging and anchoring such a network has not been elucidated (*Ma and Berro, 2018*).

Second, the effective anchoring of actin filaments to the surface of the pit depends on the distribution of linker proteins on the pit surface. Since these linker proteins are embedded within the clathrin coat (*Clarke and Royle, 2018*; *Engqvist-Goldstein et al., 2001*; *Sochacki et al., 2017*), their surface coverage is directly proportional to the coat coverage on the endocytic pit. This observation suggests that one possible function of a large coat is for the actin-linking proteins Hip1, Hip1R and Epsin to cover enough surface area to provide leverage for internalization. The role of linker protein coverage in force generation also has implications for the flat-to-curved transition earlier in endocytosis, when the membrane either begins to bend from a flat pre-assembled coat or continually deforms while the coat assembles (*Avinoam et al., 2015*; *Bucher et al., 2018*; *Scott et al., 2018*; *Sochacki and Taraska, 2019*). In cases when the clathrin coat gradually increases in area during membrane deformation, our findings imply that actin polymerization may be ineffective until the coat reaches a threshold size (*Avinoam et al., 2015*; *Sun et al., 2017*), with membrane tension controlling a switch for the flat-to-curved transition (*Bucher et al., 2018*; *Scott et al., 2018*). Future work will investigate the relationship between coat topology and actin forces during the initiation of endocytosis.

Third, we showed a significant fraction of endocytic actin filaments bend under force. We predict that the bent filaments, whose existence we confirmed by cryo-electron tomography of intact cells, contribute to successful endocytic internalization in at least two ways. First, they might contribute to the resilience of endocytosis by preventing backsliding of the pit. Second, we expect that they contribute to internalization by releasing stored elastic energy when they straighten out under thermal fluctuations, consistent with the elastic Brownian ratchet mechanism for actin-mediated force production (*De La Cruz and Gardel, 2015a*; *Mogilner and Oster, 1996*). Here, filament bending occurs to a greater extent and for a longer time than previously described for coherent flat surfaces like the

leading edge, possibly due to the curved geometry of endocytic pits. Fixing the filament orientation at one end increases bending energy (*De La Cruz et al., 2015b*; *Fernández et al., 2006*), which is accomplished here by multivalent attachments from Hip1R. Previous studies overlooked the role of actin filament bending at endocytic sites because of the predicted short length of filaments based on population averages (*Berro et al., 2010*) and the possible loss of less densely branched filaments during the preparation process for platinum-replica electron microscopy (detergent extraction or sonication-based unroofing) (*Collins et al., 2011*). The load response of branched actin networks in vitro can be reversible due to filament bending (*Chaudhuri et al., 2007*), or permanent from a change in filament architecture (*Bieling et al., 2016*; *Parekh et al., 2005*). In our simulations, some of the elastic energy from bent filaments is released as internalization increases, suggesting a reversible compression of the network to store elastic energy (*Figure 5*). However, a significant fraction of filament bending is retained after the spring is released, which suggests that load also changes the intrinsic structure of the network (*Bieling et al., 2016*).

Importantly, the results presented here demonstrate a mechanism of active adaptation by the endocytic actin network to changes in load (*Figure 7*). Different cell types, different locations in the same cell, and different stages of endocytosis at the same location can have different membrane tension values at different times (*Shi et al., 2018*). Under flat geometries, branched actin networks adapt to load (*Bieling et al., 2016*; *Mueller et al., 2017*). Here, the distribution of Hip1R linkers around the pit directs more filaments to grow toward the base of the pit (*Figure 4*), which nucleates more filaments autocatalytically and increases filament bending (*Figure 5*), thereby supporting greater internalization (*Figure 7*).

It is now important to determine whether the principles of actin filament self-organization and load adaptation identified here also apply to endocytic actin in the higher force regime characteristic of fungi. An agent-based model of endocytic actin networks in yeast predicted that barbed filament ends radiate away from the center of pit in the XY plane (*Mund et al., 2018*). However, the >200 x larger force requirements in this organism lead to a different axial organization of the filaments, less filament bending, and a distinct mechanism of force production sufficient to counteract high turgor pressure. Understanding the mechanical function and load adaptation in the 'soft' regime studied here is likely to apply to a variety of cellular membrane bending processes employing branched actin networks, including the assembly and maturation of endosomes, lysosomes, and autophagosomes (*Rottner et al., 2017*).

Finally, we acknowledge that our model represents a minimal core actin machinery necessary for endocytic internalization in mammalian cells. This feature of our study was necessary so the number of free parameters could be limited. Future models will add complexity to test roles for filament crosslinking, filament severing, and myosin-I motor activity, among other factors. It is anticipated that these additional features will increase our understanding of the force-generation capability and overall efficiency of the endocytic actin network.

## Materials and methods

### Key resources table

| Reagent type (species) or resource | Designation | Source or reference | Identifiers | Additional information |
|---|---|---|---|---|
| Gene (*H. sapiens*) | ArpC3 | | HGNC:706 ARPC3 | |
| Cell line (human) | WTC-10 hiPSC AP2-tagRFP-t ArpC3-tagGFP2 | This study | | Cell line maintained in D. Drubin lab |
| Transfected construct (*H. sapiens*) | 12mer-tagGFP2-FKBP | This study | | Plasmid to transiently express calibration construct |
| Transfected construct (*H. sapiens*) | 24mer-tagGFP2-FKBP | This study | | Plasmid to transiently express calibration construct |

*Continued on next page*

*Continued*

| Reagent type (species) or resource | Designation | Source or reference | Identifiers | Additional information |
|---|---|---|---|---|
| Transfected construct (*H. sapiens*) | 60mer-tagGFP2-FKBP | This study | | Plasmid to transiently express calibration construct |
| Transfected construct (*H. sapiens*) | 120mer-tagGFP2-FKBP | This study | | Plasmid to transiently express calibration construct |
| Antibody | mouse monoclonal anti-GAPDH | ProteinTech | 10494–1-AP (RRID:AB_2263076) | (1:5000 dilution) |
| Antibody | tag(C,G,Y)FP | Evrogen | 12101231265 | (1:2500 dilution) |
| Sequence-based reagent | ArpC3 crRNA | This paper | crRNA | CCGGGCUCCCUUCACUGUCC |
| Sequence-based reagent | ArpC3_sequencing primer | This paper | PCR primers | ACTTATTCTTATTAAGCGCCAGC |
| Sequence-based reagent | ArpC3_sequencing primer | This paper | PCR primers | CAGGGCTCTGGAGACGGT |
| Commercial assay or kit | Lipofectamine Stem | Thermo Fisher | STEM00003 | |
| Chemical compound, drug | AP21967 | Clontech | 635056 | |
| Chemical compound, drug | CK-666 | Sigma | SML0006 | |
| Software, algorithm | Cytosim | *Nedelec and Foethke, 2007* | | https://github.com/DrubinBarnes/Akamatsu_CME_manuscript |
| Software, algorithm | MATLAB | Mathworks, Inc | R2017b | |
| Software, algorithm | Python | Python.org | 3.7 | |
| Software, algorithm | µ-Track | *Jaqaman et al., 2008* | | https://github.com/DrubinBarnes/Akamatsu_CME_manuscript |

## Mathematical modeling

We combined a continuum membrane mechanics model with filament-based simulations of actin polymerization coupled to endocytic pit internalization to develop a multiscale model of actin in mammalian endocytosis. In the continuum model, the bending of the membrane was explicitly modeled and the contributions of the actin cytoskeleton were simplified to an applied localized force, while the agent-based model simplified the membrane and explicitly modeled actin and associated binding proteins (*Scheme 1*).

We iteratively ran simulations in each module. The results from each module improved the assumptions of the other, over successive rounds of simulation and comparison to experimental measurements. In particular, experiments focused our attention on the internalization of the U-shaped pit and its transition to omega shape for both the membrane and actin modules. Experiments and the actin module informed the location of actin forces for the membrane mechanics model. The membrane mechanics simulations in turn informed the initial pit shape and force/extension relationship for the actin module. Thus, these two modules synergistically provided collective information about how actin organization and dynamics couple to the bending and internalization of the clathrin-coated pit.

## Membrane mechanics module

Continuum mechanics modeling of the plasma membrane allows a quantitative understanding of the relationship between applied forces and the shape of the membrane (*Derényi et al., 2002*; *Rangamani et al., 2013*). Bending the membrane requires energy, such that pulling a tether from a flat membrane requires increasing force until the membrane adopts a tubule shape (*Derényi et al., 2002*; *Alimohamadi et al., 2018*). Adding a region with spontaneous curvature (corresponding to the endocytic coat) can help lower this energy barrier against moderate membrane tension (*Rangamani et al., 2013*; *Hassinger et al., 2017*). Forces due to actin polymerization can also help overcome the energy barrier (*Hassinger et al., 2017*), but the relationship between applied actin

forces and coated membrane shape has not been explored quantitatively. The following assumptions guide our model of the coated plasma membrane:

- Membrane curvature generated due to forces or protein-induced spontaneous curvature is much larger than the thickness of the bilayer. Based on this assumption, we model the lipid bilayer as a thin elastic shell with a bending energy given by the Helfrich-Canham energy, which is valid for radii of curvatures much larger than the thickness of the bilayer (*Helfrich, 1973*).
- We neglect the surrounding fluid flow or inertial dynamics and assume that the membrane is at mechanical equilibrium at all times (*Naghdi, 1957*; *Steigmann et al., 2003*). This assumption is commonly used in the modeling of membrane curvature to keep the mathematics tractable (*Alimohamadi et al., 2018*; *Hassinger et al., 2017*; *Rangamani et al., 2014*; *Steigmann et al., 2003*; *Vasan et al., 2018*).
- The membrane is incompressible because the energetic cost of stretching the membrane is high (*Steigmann et al., 2003*; *Steigmann, 1999*). This constraint is implemented using a Lagrange multiplier (*Alimohamadi et al., 2018*; *Rangamani et al., 2014*; *Rangamani et al., 2013*).
- Finally, for simplicity in the numerical simulations, we assume that the membrane in the region of interest is rotationally symmetric. The following derivation can also be found in *Hassinger et al. (2017)*.

We use a modified form of the Helfrich energy defined as

$$W(H, K, \theta^\alpha) = \kappa(H - C(\theta^\alpha))^2 + \bar{\kappa}K$$

A general force balance on the membrane can be written as

$$\nabla \cdot \sigma + p\mathbf{n} = \mathbf{f},$$

where is $\nabla\cdot$ surface divergence, $\sigma$ is the stress vector, $p$ is the pressure difference between the inside and outside of the volume bounded by the membrane, and $\mathbf{f}$ is any externally applied force per unit area on the membrane. The stress vector can be split into normal and tangential components as

$$\sigma^\alpha = \mathbf{T}^\alpha + S^\alpha\mathbf{n},$$

where

$$\mathbf{T}^\alpha = T^{\alpha\beta}\mathbf{a}_\beta, \quad T^{\alpha\beta} = \sigma^{\alpha\beta} + b_\mu^\beta M^{\mu\alpha}, \quad S^\alpha = -M_{;\beta}^{\alpha\beta}.$$

$\sigma^{\alpha\beta}$ and $M^{\alpha\beta}$ can also be written as

$$\sigma^{\alpha\beta} = \rho\left(\frac{\partial F(\rho, H, K; x^\alpha)}{\partial a_{\alpha\beta}} + \frac{\partial F(\rho, H, K; x^\alpha)}{\partial a_{\beta\alpha}}\right),$$
$$M^{\alpha\beta} = \frac{\rho}{2}\left(\frac{\partial F(\rho, H, K; x^\alpha)}{\partial b_{\alpha\beta}} + \frac{\partial F(\rho, H, K; x^\alpha)}{\partial b_{\beta\alpha}}\right),$$

Here $(a^{\alpha\beta}) = (a_{\alpha\beta})$ is the dual metric or first fundamental form, $b_{\alpha\beta}$ is the second fundamental form and $\rho$ is the surface mass density. Using the first and second fundamental form, we can define H (mean curvature) and K (Gaussian curvature) as

$$H = \frac{1}{2}a^{\alpha\beta}b_{\alpha\beta}, \quad K = \frac{1}{2}\varepsilon^{\alpha\beta}\varepsilon^{\lambda\mu}b_{\alpha\lambda}b_{\beta\mu}.$$

where $\varepsilon^{\alpha\beta}$ is the permutation tensor defined by $\varepsilon^{12} = -\varepsilon^{21} = \frac{1}{\sqrt{a}}, \varepsilon^{11} = \varepsilon^{22} = 0$.

We then define an area incompressibility constraint by rewriting the free energy density as

$$F(\rho, H, K; x^\alpha) = \tilde{F}(H, K; x^\alpha) - \frac{\gamma(x^\alpha, t)}{\rho}.$$

where $\gamma(x^\alpha, t)$ is a Lagrange multiplier field required to impose invariance of $\rho$ on the whole of the surface. This free energy density relates to the Helfrich energy density as

$$W = \rho \tilde{F}$$

Combining these equations, we can get the stress equations

$$\sigma^{\alpha\beta} = (\lambda + W)a^{\alpha\beta} - (2HW_H + 2\kappa W_K)a^{\alpha\beta} + W_H \tilde{b}^{\alpha\beta},$$

and

$$M^{\alpha\beta} = \frac{1}{2}W_H a^{\alpha\beta} + W_\kappa \tilde{b}^{\alpha\beta},$$

where $\lambda = -(\gamma + W)$ Simplifying this further, we can get the shape equation (normal balance)

$$p + \mathbf{f} \cdot \mathbf{n} = \Delta \frac{1}{2}W_H + (W_K);_{\alpha\beta}\tilde{b}^{\alpha\beta} + W_H(2H^2 - K) + 2H(KW_K - W) - 2\lambda H,$$

and the tangential balance

$$\left( \frac{\partial W}{\partial x^\alpha_{|exp}} + \lambda, \alpha \right)a^{\beta\alpha} = \mathbf{f}.\mathbf{a}_s.$$

where $()_{|exp}$ denotes the explicit derivative respect to coordinate $\theta^\alpha$. To further simplify the equations, we define the coordinate system as axisymmetric using

$$\mathbf{r}(s, \theta) = r(s)\mathbf{e}_r(\theta) + z(s)\mathbf{k}.$$

We define a $\psi$ such that $r'(s) = \cos(\psi)$, $z'(s) = \sin(\psi)$ and $\mathbf{n} = -\sin\psi\mathbf{e}_r(\theta) + \cos\psi\mathbf{k}$
Using this, we can write the mean curvature (H) and Gaussian curvature (K) as

$$H = \frac{1}{2}(\kappa_\nu + \kappa_\tau) = \frac{1}{2}(\psi' + r^{-1}\sin\psi)$$

$$K = \kappa_\tau \kappa_\nu = \frac{\psi'\sin\psi}{r}.$$

We also introduce

$$L = \frac{1}{2\kappa}r(W_H)'$$

allowing us to formulate a system of ordinary differential equations (ODE's) as function of arc length s

$$
\begin{aligned}
r' &= \cos\psi, \\
z' &= \sin\psi, \\
r\psi' &= 2rH - r' = -\sin\psi, \\
rH' &= L + rC', \\
\frac{L'}{r} &= \frac{p}{k} + \frac{\mathbf{f}\cdot\mathbf{n}}{\kappa} + 2H\left[(H-C)^2 + \frac{\lambda}{\kappa} - 2(H-C)\right]\left[H^2 + (H - r^{-1}\sin\psi)^2\right] \\
\lambda' &= 2\kappa(H-C)C' - \mathbf{f}\cdot\mathbf{a}_s.
\end{aligned}
$$

This can also be written as a function of membrane area using

$$a(s) = 2\pi\int_0^s r(\xi)d\xi \rightarrow \frac{da}{ds} = 2\pi r.$$

Here, we choose to non-dimensionalize the system using

$$\alpha = \frac{a}{2\pi R_0^2}, \; x = \frac{r}{R_0}, \; y = \frac{y}{R_0}, \; h = HR_0, c = CR_0, \; l = LR_0$$

$$\lambda^* = \frac{\lambda R_0^2}{\kappa_0}, p^* = \frac{pR_0^3}{\kappa_0} f^* = \frac{fR_0^3}{\kappa_0}, \; \kappa^* = \frac{\kappa}{\kappa_0}$$

giving us the system of equations

$$
\begin{aligned}
x\dot{x} &= \cos\psi, \\
x\dot{y} &= \sin\psi, \\
x^2\dot{\psi} &= 2xh - \sin\psi, \\
x^2\dot{h} &= l + x^2\dot{c}, \\
\dot{l} &= \tfrac{p^*}{\kappa^*} + \tfrac{\mathbf{f}^*\cdot\mathbf{n}}{\kappa^*} + 2h\left[(h-c)^2 + \tfrac{\lambda^*}{\kappa^*}\right] - 2(h-c)\left[h^2 + (h - x^{-1}\sin\psi)^2\right], \\
\dot{\lambda}^* &= 2\kappa^*(h-c)\dot{c} - \tfrac{\mathbf{f}^*\cdot\mathbf{a}_s}{x}.
\end{aligned}
$$

We define a spatially varying spontaneous curvature as

$$
c = c_0 * 0.5(1 - \tanh(g*(a - a_0)))
$$

where $a$ is the non-dimensional membrane area, $a_0$ is the non-dimensional membrane area of the protein coat, $g$ is a constant and $c_0$ is the coat spontaneous curvature. The parameters used for the spontaneous curvature simulations are specified in *Supplementary file 1*:

To perform the coat pulling simulations, we applied an axial force acting downward along the protein coat and upward along the base of the pit such that the net force integrates to 0 (we do this by scaling the applied force by the area over which it is applied). This force function was defined as

$$
\mathbf{f} = \mathbf{f}_0 * (0.5((1 - \tanh(g*(a - a_0)))/a_0 - (tanh(g*(a - a_{in})) - tanh(g*(a - a_{out})))/(a_{out} - a_{in})))
$$

where $a_0$ is the non-dimensional coat area, $a$ is the non−dimensional membrane area, $a_{out} - a_{in}$ is the area of force applied at the base of the pit. $a_{in}$ corresponds to an inner radius $r_{in}$ and $a_{out}$ corresponds to an outer radius $r_{out}$ within which the upward force is applied.

These parameters are specified in *Supplementary file 2*.

To simulate the pinched ('omega-shaped'") curves at high membrane tension, we provided an initial guess of an 'omega-shaped' membrane from a lower membrane tension. We did this because the simulations stalled at U shapes at an internalization of about 100 nm. Providing this initial guess led to solutions for membrane shape and force beyond 100 nm, as seen in *Figure 1C*. Further, to fully explore the space of solutions, we ran the simulations backward by starting from an 'omega shaped' pit at a large internalization and then decreasing the internalization. In *Figure 1C*, we plotted the curves for negative internalization > = the farthest internalization for the U shaped pit (generally ~100 nm). Values of membrane tension in *Figure 1C* are [0:051:0.05:0.451] pN/nm, and rounded to two significant digits in the figure for clarity.

## Actin module

We used Cytosim (*Nedelec and Foethke, 2007*) to model the polymerization of a branched actin network coupled to the internalization of a clathrin-coated pit. This approach simplified the pit as a bead attached to a flat boundary (the plasma membrane) by a spring. This assumption of a linear force-extension relationship was validated in *Figure 1*. Actin filaments and actin-binding proteins (Arp2/3 complex, Hip1R) were explicitly simulated as individual objects (agents). Cytosim calculates the forces on each segment of actin from rules such as diffusion, confinement, growth, and binding based on Brownian dynamics.

### Assumptions in Cytosim

Cytosim simulates the movement of actin model points within a boundary according to constrained Langevin dynamics (*Nedelec and Foethke, 2007*), which accounts for the diffusion, bending, and forces of actin filaments, as well as the diffusion and binding of actin-binding proteins detailed below.

1. Force balance equations (*Nedelec and Foethke, 2007*) section 7.1:
    a. All points in the simulation follow constrained Langevin dynamics:
    b. $dx = \mu F(x, t)dt + dB(t)$
        i. where μ is defined as an effective mobility, which takes on a different value for each type of object.
2. Mobilities of diffusing objects:
    a. The simulated Brownian motion $dB(t)$ of objects of radius r is governed by a uniformly distributed random $[0, 1] * \sqrt{k_B T/3\pi\eta}$ number at each time point, where $\eta$ is the viscosity of the cytoplasm.

 b. Their movement is governed by a mobility μ
 i. For model points of an actin filament, $\mu = \log(L/\delta)/(3\pi\eta L)$ for a rod of diameter $\delta$, length L and cytoplasmic viscosity $\eta$. This mobility term ignores the bending of the filaments.
 c. The endocytic pit is modeled as a solid, with bulk fluid viscosity associated with pit translational movement and a viscoelastic confinement to the cell surface: $dx = \mu F(x,t)dt + dB(t)$, where $\mu = 6\pi\eta r$.

3. Confinement of objects:
 a. Objects are confined within a boundary (cell surface) according to a harmonic spring potential $F = kx$.
 i. The endocytic pit a distance $z$ from the cell surface experiences a force $F = kz$.
 ii. Actin filaments are confined inside the cell wherein each model point at distance $z$ outside the cell experiences a force $F = kz$.

4. Bending elasticity of filament model points:
 a. Filament model points are connected via linear elasticity according to a flexural rigidity $\kappa$, which is the persistence length $L_p$ multiplied by $k_B T$.
 b. The bending elasticity is treated as linear (see Limitations) such that for three connected actin model points $m_0, m_1, m_2$ the force for those points is $F = \kappa(p/L)^3(m_0 - 2m_1 + m_2)$, where $\kappa$ is the flexural rigidity, $p$ is the number of model points, and $L$ is the length of the filament.

5. Actin-binding proteins

 a. Hip1R binds actin filaments according to a binding rate and binding radius (probability of binding when a filament is within the binding radius). This general actin-binding protein is a simplification of the multiple interacting proteins that link actin to the coat, including Hip1 and Epsin1/2/3 (*Brett et al., 2006*; *Chen et al., 1998*; *Messa et al., 2014*; *Senetar et al., 2004*; *Skruzny et al., 2012*).
 b. Arp2/3 complex was developed as a special-case 'fork' class with two coupled ends. One end binds actin filaments, and the other nucleates a new actin filament, provided the first end bound an actin filament (this is defined as trans-activation in Cytosim). In the 'fork' class, the two ends are constrained geometrically at a resting angle with a given resistance to torque (angular stiffness) similar to 4b above (*Mund et al., 2018*).

## Assumptions for modeling mammalian clathrin-mediated endocytosis in cytosim

### Geometry

Endocytic pit: We used our membrane mechanics simulations (*Figure 1*; *Hassinger et al., 2017*) to estimate the dimensions of the endocytic pit for physiological values of membrane tension and rigidity of the membrane and clathrin coat. Under these conditions the clathrin coat bends the plasma membrane into a U-shaped hemisphere (*Figure 1*; *Boulant et al., 2011*; *Messa et al., 2014*; *Yarar et al., 2005*). We initialized the pit as a hemisphere 90 nm in diameter (*Avinoam et al., 2015*; *Collins et al., 2011*). As the pit internalizes, a smaller neck is exposed (*Figure 1*), which is modeled as a sphere with a cylindrical neck of diameter 60 nm. Internalization is defined as a displacement in the -Z direction (*Figure 1A*).

 Active Arp2/3 complex: We collapsed the activation steps of Arp2/3 complex into a single species, active Arp2/3 complex, that resides on the plasma membrane from the beginning of the simulation. This models the cellular process, in which soluble Arp2/3 complex is inactive until it encounters its activator N-WASP at the plasma membrane.

 N-WASP binds the plasma membrane via a PI(4,5)P2-binding site (which relieves its own autoinhibition) (*Rohatgi et al., 2000*). Additional proteins can bind different regions of N-WASP to increase its level of activation, including the GTPase Cdc42, actin nucleator cortactin, and BAR protein SNX9. Because the activation rate and concentrations of these proteins are not yet known, we considered fully active N-WASP (similar to the VCA region alone) rather than modeling the individual activation steps. Furthermore, rather than explicitly modeling N-WASP, we used pre-activated Arp2/3 complex, which models the coincidence of active N-WASP with soluble Arp2/3 complex on the plasma membrane. This active Arp2/3 complex can template new branched actin filaments when in proximity of an existing 'mother' actin filament. Thus, this model aims to functionally capture Arp2/3

complex activation and the geometry of branched actin filament nucleation, rather than explicitly modeling each molecule involved in the process of Arp2/3 complex activation.

N-WASP (or its homologues Las17/WASP in yeast) accumulates earlier in endocytosis (*Taylor et al., 2011*) until a 'threshold' concentration triggers actin assembly (*Sun et al., 2017*); here we initialize all the active Arp2/3 complex on the plasma membrane at the beginning of the simulation, and it is used over the course of the simulation. Therefore we model the phase in which a threshold value of Arp2/3 complex activators has accumulated at the endocytic site and is ready to trigger actin polymerization.

We assumed that activated Arp2/3 complex resides in a ring around the base of the endocytic pit. This feature has been shown for the budding yeast homologue of N-WASP, Las17 (*Picco et al., 2015*). Endocytic actin polymerizes from the base of the pit in budding yeast (*Idrissi et al., 2008*; *Kaksonen et al., 2003*; *Mund et al., 2018*; *Picco et al., 2015*) and in mammalian cells (Kaplan et al., in preparation; *Almeida-Souza et al., 2018*), consistent with active Arp2/3 complex residing in a ring at the base of the pit. In our fluorescence micrographs, Arp2/3 complex is diffraction-limited, so the outer diameter of this ring is ≤250 nm. The inner diameter of the ring of Arp2/3 complex corresponds to the width of the neck of the pit, 60 nm. In budding yeast the Las17 ring outer diameter is ~140 nm (*Mund et al., 2018*), which corresponds to a surface density of ~3000 molecules/$\mu m^2$. We conservatively set the outer radius of the ring to be 240 nm, which also corresponds to a surface density of ~3000 molecules/$\mu m^2$. Estimates of the surface density of in vitro and in vivo patterned activators of Arp2/3 complex (also called nucleation-promoting factors) range from ~3000–19000 molecules/$\mu m^2$ (*Bieling et al., 2018*; *Case et al., 2019*; *Ditlev et al., 2012*).

## Filament attachments to endocytic pit

In mammalian cells, Hip1R and Epsin connect actin filaments to the endocytic pit (*Brett et al., 2006*; *Chen et al., 1998*; *Engqvist-Goldstein et al., 2001*; *Messa et al., 2014*; *Senetar et al., 2004*; *Skruzny et al., 2012*). Both are present throughout the clathrin coat (*Clarke and Royle, 2018*; *Sochacki et al., 2017*). We wrote a script in Matlab to uniformly distribute Hip1R molecules around a desired coverage of a sphere, from 1% to 80% of a sphere (*Figure 4*). In most simulations Hip1R was distributed in 60% of a sphere.

## Modeling actin filament dynamics

Stall force: Filament polymerization slows under applied load according to the Brownian Ratchet theory (*Peskin et al., 1993*). This is treated in Cytosim as growth velocity that slows exponentially under applied load, which is reasonable within the time scales of endocytosis.

## Modeling filament capping

Previous filament-based models of actin in endocytosis modeled actin filaments with uniform lengths or that grow until a maximum length (*Mund et al., 2018*), while others took into account stochastic capping without diffusion or bending (*Wang et al., 2016*). We adapted an existing property in Cytosim to model stochastic filament capping, such that the capping events were exponentially distributed. We modeled actin filaments using Cytosim's 'classic' fiber class, which treats the filament as growing from its plus end, with a stochastic probability of depolymerizing (corresponding to catastrophe for microtubules). We set the depolymerization rate of shrinking filaments to be 0 with no recovery rate. Thus these filaments become irreversibly capped after an exponential wait time characterized by the parameter catastrophe_rate (which we define as the capping rate). Because the probability that capping protein binds to the barbed end of the filament is exponentially distributed, filament lengths are exponentially distributed (*Figure 3—figure supplement 1F*). We set the rate of capping to achieve a desired mean filament length based on the expected distributions of actin filament lengths for a given capping rate. In Cytosim, the catastrophe rate can be set to depend on a combination of applied load and growth velocity, which we did not include in our model of actin filament capping.

## Source of actin mother filaments

Active Arp2/3 complex requires a mother filament from which to nucleate a new actin filament at a defined angle (*Amann and Pollard, 2001*; *Mullins et al., 1997*). Therefore, the polarity of the

mother filament defines the polarity of the resultant branched actin network. Our study uses diffusing linear actin filament nucleating proteins (*Balzer et al., 2018*; *Basu and Chang, 2011*; *Wagner et al., 2013*) to seed a defined number of randomly oriented mother filaments near the endocytic site. Alternatively, simulations using a pool of cytoplasmic linear actin filaments (*Raz-Ben Aroush et al., 2017*) allowed for similar internalization, but with less reliable timing of initiation. More detailed studies of the mechanism of actin nucleation and mother filament generation are necessary.

## Limitations

Endocytic pit internalization is simplified as an elastic spring, with a linear relationship between force and extension. We show in *Figure 1* that this linear relationship is characteristic of coated plasma membranes under force up until a threshold internalization of ~100 nm. Future studies will treat the coated membrane as a 3D, force-dependent curving surface; such an approach is outside the scope of the present work.

We focus our model on the minimal actin machinery required to produce force. We have not included crosslinkers (*Ma and Berro, 2018*) or myosin I, both of which are expected to increase the network's ability to produce force. The effects of these two proteins on mammalian endocytosis will be treated in a future study.

The treatment of filament bending elasticity as linear is valid for small deflections of individual actin model points. Importantly, the outcomes of our simulations did not depend on the frequency of segmentation of actin model points (which change the magnitude of deflection between individual actin model points). Filament twist or twist-bend coupling, which increases the total energy stored in bent actin filaments (*De La Cruz et al., 2015b*), is not considered in Cytosim, and requires a more detailed modeling approach considering each subunit. Cytosim does not implement hydrodynamics for curved filaments, so the diffusion of these filaments is approximated as the motion for a linear filament.

Arp2/3 complex preferentially binds to the curved sides of actin filaments (*Risca et al., 2012*). We do not include this assumption in our model. We expect that the self-organization and robustness exhibited by our minimal actin network would be enhanced by this assumption, given that Arp2/3 complex at the base of the pit encounters many curved sides of actin filaments.

## Parameter values

We derived most parameters from experimental data in the literature, and made measurements for some measurements not available (*Supplementary file 3*). We varied the remaining parameters to show their effect on the outcome of the simulations. Discussion of each parameter follows below.

## Membrane tension

We used the relationship between internalization resistance and membrane tension (*Figure 1*) to calibrate the spring stiffness in our agent-based simulations. We relied on values of membrane tension measured in human skin melanoma SK-MEL-2 cells, based on atomic force microscope membrane tether rupture forces and the assumption that the rigidity of the plasma membrane is 320 pN·nm (*Dimova, 2014*; *Diz-Muñoz et al., 2016*; Kaplan et al., in preparation).

## Association rate constants

The biochemical association rate constant, $k_{on}$, is given in units of $\mu M^{-1}\ s^{-1}$. In Cytosim, we input the association probability between actin and a binding protein as a binding_rate, in units of $s^{-1}$. These two rates can be related by the following relationship:

$k_{on} = binding\_rate * capture\_volume$, which is defined as $\pi * capture\_radius^2 * filament\_length$ (Francois Nedelec, personal communication). This gives an order-of-magnitude scaling relationship to convert between $k_{on}$ and binding rate, considering that cytosim does not treat explicit binding sites on the filament (Francois Nedelec, personal communication).

## Arp2/3 complex

Branching angle: Based on *Blanchoin et al. (2000)* we set the branching angle of Arp2/3 complex to be 77 ± 13°, as measured for bovine Arp2/3 complex. Acanthomeba Arp2/3 complex adopts closer

to a 70° branching angle. In NIH3T3 cells preserved by cryo-fixation, branch angles in the lamellipodium are 77 ± 8° (*Vinzenz et al., 2012*). Importantly, *Blanchoin et al. (2000)* was the only in vitro study we are aware of that measured the variance of branching angles, which was converted to an angular stiffness (0.076 pN/rad$^2$). Therefore we set the resting angle of Arp2/3 branches as 1.344 rad with an angular stiffness of 0.076 pN/rad$^2$.

## Unbinding rate

We used the $k_{off}$ measured by *Beltzner and Pollard (2008)*: 0.003s$^{-1}$.

## Nucleation rate

Based on the association rate constant between activated Arp2/3 complex and actin filaments of 0.00015 µM$^{-1}$ s$^{-1}$ (*Beltzner and Pollard, 2008*), we set the binding_rate of Arp2/3 to actin to be 7 s$^{-1}$, given a capture radius of 10 nm and filament length of 100 nm (see relationship between these parameters above). We note that this calculation of *binding_rate* depends inversely with the filament length (and number of filaments) which are not directly comparable in our simulations given that the filaments and Arp2/3 are generally not freely diffusing. Still, it was remarkable that our best estimate for *binding_rate* gave reasonable nucleation kinetics, and served as a threshold for timely internalization of the endocytic pit, whereas previous deterministic models needed to increase the association rate constant by 300-600x for efficient nucleation (*Beltzner and Pollard, 2008*; *Berro et al., 2010*). These ODE models did not have spatial considerations, so this suggests that the spatial and temporal confinement of actin filaments and the high local concentration of active Arp2/3 complex in our simulations accounted for most of this difference. Thus the local geometry has a significant (>2 orders of magnitude) effect on the effective nucleation rate.

## Actin

Growth rate: In cells the cytoplasmic concentration of actin is 60–100 µM (*Haugwitz et al., 1994*; *Wu and Pollard, 2005*). In mammalian cells, a subset of this actin is available as polymerizable actin, both due to monomer-sequestering proteins (thymosin B4) and due to only a subset of monomers being ATP-bound. We conservatively set the concentration of available polymerizable actin to be 20 µM, which given the association rate constant of ATP-actin of 11.6 subunits/µM/s (*Pollard, 1986*) corresponds to a polymerization rate of 500 nm (182 subunits) per second.

Capping rate: The mean length of filaments in mammalian endocytosis has not been measured. We relied on the estimates from *Berro et al. (2010)*; *Sirotkin et al. (2010)* which showed that for fission yeast filaments were an average of 150 nm in length. We set the capping rate to be 6.3/s, which set the mean filament length at 150 nm. We varied the rate of capping in our simulations. Less actin capping resulted in greater internalization, due to more actin (*Figure 1—figure supplement 1*). However, the resultant amount of actin is larger than the amount of actin measured in CME in other organisms (*Picco et al., 2015*; *Sirotkin et al., 2010*).

Stall force: The stall force scales with the load applied and the concentration of actin monomers available (*Peskin et al., 1993*). At 4 µM actin the filaments' stall force was measured to be 1–2 pN (*Footer et al., 2007*). With 40 µM actin the filaments could theoretically stall at up to 9 pN force per filament (*Dmitrieff and Nédélec, 2016*). For the ~20 µM actin that we assumed was available for polymerization, the stall force was ~5 pN. Surprisingly, the extent of internalization varied weakly with stall force (*Figure 1—figure supplement 1*), suggesting that actin used another mode of force generation than elongation directly against the membrane (*Figure 5*).

Persistence length: We set the persistence length of actin filaments to be 10 µm, which corresponds to a flexural rigidity of 0.041 pN·µm$^2$ (*McCullough et al., 2008*). Previous modeling studies used a value of 20 µm, based on measurements of actin filaments labeled with phalloidin (*Gittes et al., 1993*), which stiffens actin filaments (*Isambert et al., 1995*; *Pfaendtner et al., 2010*). Changing the persistence length of actin between 1 and 20 µm had a minor effect on pit internalization (*Figure 1—figure supplement 1*).

## Hip1R

In mammalian endocytosis, several proteins link actin filaments to the clathrin coat via phosphoinositide- and clathrin-binding domains and actin-binding domains, including Hip1, Hip1R, and Epsin

(*Brett et al., 2006*; *Chen et al., 1998*; *Messa et al., 2014*; *Senetar et al., 2004*; *Skruzny et al., 2012*). Our general linker protein, named in the text as Hip1R, is a surrogate for all proteins that link actin to the coat.

Endocytic actin-binding proteins Hip1 and Hip1R use a conserved domain to bind actin with surprisingly weak affinity. This domain, which is alternately named the THATCH (Talin- Hip1/R/Sla2p Actin-Tethering C-terminal Homology), Talin-family, or I/LWEQ domain, has been isolated and studied by several groups. We fit the binding results of previous Hip1R binding experiments in our lab (*Engqvist-Goldstein et al., 1999*) to estimate a binding affinity between Hip1R and actin as ~400 nM, and the affinity between clathrin cages and Hip1R to be ~1 nM. Both sets of data were fit well by the quadratic binding curve (*Pollard, 2010*),

$$[LR]/[L] = (([R] + [L] + K_d) - (([R] + [L] + K_d)^2 - 4 * [R] * [L])^{1/2}/(2 * [L])$$

where $[L]$ is the concentration of actin or clathrin and $[R]$ is the concentration of Hip1R, with $r^2 = 0.94$ and $0.99$, respectively (data not shown). Other studies measured a weaker affinity between Hip1R and actin: $K_d = 3.4$ µM, or 2.5 µM for Hip1 (*Senetar et al., 2004*). In the presence of the first helix of the five-helix bundle comprising the THATCH domain, actin affinity further decreases (*Senetar et al., 2004*). Epsin has two actin-binding domains with very weak ($K_d > 10$ µM) or unknown affinity to actin (*Messa et al., 2014*; *Skruzny et al., 2012*). For our linker protein we used a combination of rate constants such that $k_{off}/k_{on} \sim K_d$ of 400 nM. Compared with dilute reactions, in an endocytic geometry actin filaments grow near the coat, so the actin filaments bind Hip1R more frequently. As a result we found that a relatively low binding rate was sufficient for binding between actin and Hip1R in our simulations. We varied the off rate of Hip1R and found that that, surprisingly, the internalization was robust to Hip1R off rate between 0.001 and 10 s$^{-1}$ (*Figure 1—figure supplement 1*).

With such weak affinity, $\geq 100$ linking molecules are required for robust endocytosis (*Figure 4*). The following considerations support the likelihood that a sufficient number of actin-linking proteins reside in the clathrin-coated pit. In yeast, endocytic sites accumulate ~100–150 molecules of the Hip1R homologue End4p and 230 molecules of the actin-binding protein Pan1 (with $K_d$ to actin = 2.9 µM). Estimation from platinum replica images of clathrin-coated pits in SK-MEL-2 cells (*Sochacki et al., 2017*) suggest that clathrin cages have approximately $55 \pm 12$ 'faces' (pentagons and hexagons), or up to 90 faces in HeLa cells. If the cage accumulates one Hip1R dimer per face, this would lead to 110–180 molecules of Hip1R, plus molecules of Hip1, Epsin1, Epsin2, and Epsin3. From a similar analysis, SK-MEL-2 cells have ~$66 \pm 12$ vertices (triskelia), which corresponds to ~$200 \pm 40$ clathrin heavy chains and ~$200 \pm 40$ light chains. Since Hip1R binds clathrin light chain, a 1:1 ratio of these proteins would again suggest ~200 Hip1R molecules in the clathrin coat. Additionally, actin binding by these linker proteins is likely highly multivalent. Mammalian Epsin proteins hexamerize in vitro via their membrane-binding ENTH domains, and Hip1 and Hip1R can dimerize with each Epsin through its ANTH domain by sharing a Pi(4,5)P2 molecule (*Garcia-Alai et al., 2018*; *Skruzny et al., 2015*). Adding an additional layer of multivalency, Hip1R and Hip1 hetero- and homodimerize via coiled-coil domains and a C-terminal dimerization motif (*Brett et al., 2006*; *Chen and Brodsky, 2005*; *Engqvist-Goldstein et al., 2001*; *Niu and Ybe, 2008*; *Senetar et al., 2004*). Therefore, it is quite likely that a sufficient number of actin-linking proteins cover the coat.

## Simulation environment parameters

The internalization of endocytic pits was not sensitive to other simulation parameters including the segmentation of actin (1 and 100 nm per model point; 1 µm model points introduced additional variability), confinement force for actin within the cell, and time step of the simulation (*Figure 1—figure supplement 1*). The viscosity of the cytoplasm and the endocytic pit weakly affected extent of internalization (*Figure 1—figure supplement 1*).

## Modifications to source code

We added a method 'confine_first_surface' in which only the first segment of the multi-point bead is under confinement, and that the bead does not undergo angular displacements.

We added data reporting methods, including (1) reporting the Arp2/3 complex branch angle (Francois Nedelec); (2) reporting the Hip1R-attached filament ID numbers; (3) visualizing the axial

orientation of actin segments with respect to the plasma membrane (implemented as 75% of the hsv colormap).

## Comparison to theory

Our calculation of the elastic energy stored in bent actin filaments is derived from the theory of deforming elastic beams (Boal). Specifically, the bending energy E is determined by $(k_B T L_p \theta^2)/(2l)$, where $k_B$ is the Boltzmann constant, $T$ is temperature, $L_p$ is the filament persistence length (10 μm for actin), $\theta$ is the bending angle, and $l$ the free filament length (contour length of the filament between the attachment site and barbed end of the filament) (Boal, equations 3.15 and 3.21).

In *Figure 5* we reference (*Sept and McCammon, 2001*) and (*De La Cruz et al., 2015b*) to estimate the bending energy associated with fragmentation of bare actin filaments. Based on the energy associated with removal of two longitudinal contacts and one lateral contact between monomers in an actin filament (*Sept and McCammon, 2001*), the elastic energy associated with fragmenting an actin filament is 26–28 $k_B T$ (*De La Cruz et al., 2015b*). The rate constants in *De La Cruz and Gardel (2015a)* associated with the probability of severing are not considered in this study.

## Running simulations

We wrote custom scripts in bash to run parallel simulations on a high-performance computing server.

## Analysis of simulations

We wrote custom code in Python (3.7) with Jupyter Notebook (Project Jupyter) to read, analyze, and plot the simulations obtained from Cytosim. X,Y = 0 is defined as the center of the pit. Energy associated with polymerization was defined as 5 pN * 2.75 nm = 13.5 pN·nm per binding event. This code is available at the following website: https://github.com/DrubinBarnes/Akamatsu_CME_manuscript (*Akamatsu, 2019*; copy archived at https://github.com/elifesciences-publications/Akamatsu_CME_manuscript).

## Internalization energy

We used the relationship $E = 1/2kx^2$ to estimate the expected internalization for a non-adapting machine. From the internalization for k = 0.01 pN/nm (low tension), we calculated the expected internalization using the same energy for different values of k. Specifically, for a single value of load, we calculated the work output based on energy (work output) $E = 1/2kx^2$, where k is the load (in pN/nm) and x is the internalization (in nm). For an adapting network, the work output changes with load. For a non-adapting network, it does not, so as load k increases, internalization x decreases. This relationship of x as a function of k for constant E is plotted as the dotted line in *Figure 7E*. Energy efficiency is defined as the energy (of internalization or stored in bent filaments) divided by the total polymerization energy.

Radial orientation metric: We defined the radial orientation as the sum of the dot products of the direction of the barbed end with the unit vector in X and Y, such that −1 = barbed ends oriented toward the origin (the center of the pit), 0 = oriented tangent to the pit, and +1 = oriented away from the center of the pit.

95% internalization is defined as the 95th percentile of internalization. We use the first 12 s of simulations to allow for comparison with simulations that terminated after t = 12 s. For *Figure 4—figure supplement 2*, we compared 95% internalization in two conditions using the Welch's t-test in Python (3.7) with scipy.stats.ttest_ind(equal_var = False).

Barbed ends near base/neck: We chose a distance of 7.5 nm as a metric for proximity to the membrane (base or neck) as this was the width of 1–2 actin monomers and less than the radial distance between the neck surface and the pit diameter. The absolute value of this metric did not change the results appreciably.

## Experimental method details

### Cell line maintenance

WTC-10 human induced pluripotent stem cells (hiPSCs) were obtained from the lab of Bruce Conklin and genome edited using TALENs to endogenously express AP2-tagRFP-T at one allele at an

internal loop of the µ2 subunit (*Hong et al., 2015*). We grew these cells on matrigel (hESC-Qualified Matrix, Corning) (80 µg/mL, 1 mL/well) in StemFlex (Thermo Fisher) with Penicillin/Streptomycin (Thermo Fisher), and passaged with Gentle Cell Dissociation reagent (EDTA-based clump passaging; StemCell Technologies). Parental and genome-edited cells were tested for mycoplasma and authenticated by STR profiling. For single-cell applications (genome editing, flow cytometry, transfections) we trypsinized the cells with the recombinant trypsin TrypLE Select (Thermo Fisher) and grew the cells in StemFlex supplemented with the specific rho kinase inhibitor RevitaCell (Thermo Fisher).

## SK-MEL-2 cell culture

SK-MEL-2 cells endogenously expressing clathrin light chain CLTA-RFP and dynamin2-eGFP (*Doyon et al., 2011*) were cultured in DMEM/F12 (Thermo Fisher) supplemented with 10% FBS (HyClone) and Penicillin/Streptomycin (Thermo Fisher). A day before imaging, the cells were seeded on 25 mm diameter glass coverslips (Fisher Scientific).

## Cell line construction

We followed (*Dambournet et al., 2014*) to create lines human induced pluripotent (hiPS) cells with genetically encoded fluorescent fusion proteins. To the AP2-RFP cells described above, we used the following Cas9 ribonuclear protein electroporation method for genome editing. Gibson assembly of DNA G-blocks (IDT) was used to construct a donor plasmid containing the tagGFP2 gene, codon optimized for mammalian cell expression, between 500 base pair homology arms flanking the 3′ terminus of the ArpC3 gene. *S. pyogenes* NLS-Cas9 was purified in the University of California Berkeley QB3 MacroLab and frozen at −80 °C until use. TracrRNA and crRNAs were purchased from IDT. Cells were trypsinized in TrypLE select (Thermo Fisher), mixed with donor plasmid (final concentration 3.8 µM) and 240 pmol Cas9 (final concentration 1.75 µM) complexed with 100 µM crRNA and tracrRNA (final concentration 3.7 µM), and electroporated with an Amaxa nucleofector in stem cell nucleofector reagent (Lonza). The sequence of the crRNA to target the C terminus of the ArpC3 gene was CCGGGCUCCCUUCACUGUCC. Cells were seeded on matrigel-coated 6-well plates in StemFlex supplemented with the rho-kinase inhibitor RevitaCell and media was changed 24 hr later. Three days after electroporation the cells were bulk sorted for GFP fluorescence with a BD Bioscience Influx sorter (BD Bioscience). Fluorescence intensities of cell populations were analyzed using the flow cytometry software FlowJo (FlowJo, LLC). Around one week later the cells were re-sorted into matrigel-coated 96-well plates at densities of 5, 10, or 20 cells per well. Positive clones were confirmed by PCR and sequencing of the genomic DNA locus. From genomic DNA and fluorescent cell sorting analysis we determined that both alleles of ArpC3 were tagged with tagGFP2. We sequenced genomic DNA extracts from the cell lines at the insertion sites to confirm that there were no insertions or deletions at the site of GFP insertion. We isolated genomic DNA using the DNEasy blood and tissue DNA isolation kit (Qiagen). The primers used for PCR amplification of genomic DNA were TCAGGGTGGCTTTCTCTCCT and CCAGAGCTGCAACCAGTACA. The primers used for sequencing the ArpC3 allele were ACTTATTCTTATTAAGCGCCAGC and CAGGGCTCTGGAGACGGT.

## Western blotting

We pelleted 1–2 wells of cells from a 6-well plate (~$10^6$ cells) at 4 °C and lysed the cells in 50 mM Hepes pH 7.4, 150 mM NaCl, 1 mM MgCl$_2$, 0.1% Triton X-100, and cOmplete Mini EDTA-free protease inhibitor (Sigma Aldrich). Extracts were separated electrophoretically in sample buffer with 80 mM DTT on 8% polyacrylamide gels and transferred to nitrocellulose membranes. Membranes were blocked with 5% nonfat milk in PBS, probed with mouse monoclonal anti-GAPDH (ProteinTech 10494–1-AP) at 1:5000 dilution and rabbit polyclonal antibody against tag(C,G,Y)FP (Evrogen AB121 Lot 12101231265) at 1:2500 dilution in PBS with 0.1% TWEEN-20 and 1:100 PBS/milk, followed by incubation in donkey anti-mouse CF680 and donkey anti-rabbit CF800 (Li-Cor Biosences). Washed membranes were imaged on a Li-Cor Odyssey Clx infrared fluorescence imager (Li-Cor Biosciences).

## Choice of subunit to tag:

We chose the location of the GFP tag based on available structural, biochemical, cell biological, and genetic data on the functionality of fluorescent fusion proteins of subunits of the Arp2/3 complex.

ArpC3 (p21; Arc18 in budding yeast) tags are more functional than tags on Arp2, Arp3, or ArpC5 (Arc15 in budding yeast) (*Egile et al., 2005*; *Sirotkin et al., 2010*; *Smith et al., 2013*; *Picco et al., 2015*). Single-particle electron microscopy reconstructions and the crystal structure (PDB: 2p9l) of Arp2/3 complex show that the C terminus of ArpC3 is flexible and does not sterically interfere with Arp2/3 complex's binding site to actin filaments or its activators (VCA).

## Constructs for intracellular fluorescence-based standard curve

We adapted self-assembling protein nanocages of defined copy number (*Hsia et al., 2016*) to construct a fluorescence calibration standard in live mammalian cells. These trimeric proteins were engineered by *Hsia et al. (2016)* at protein-protein interfaces to self-assemble into a 60mer (KPDG aldolase) that can be tagged at the N or and C termini with GFP to yield an average of 60 or 120 copies of GFP. Alternatively a two-component 24mer (alamin adenosyl transferase and 5-carboxy-methyl-2-hydroxymuconate isomerase [*King et al., 2014*]) can be tagged one or both components with GFP to yield an average 12 or 24 copies of GFP per structure (*Hsia et al., 2016*). DNA constructs codon-optimized for mammalian expression were synthesized (IDT) with alanine mutations at K129 (KPDG aldolase) and R72 (5-carboxymethyl-2-hydroxymuconate isomerase) to abolish enzymatic activity. An E126L mutation in the transferase (*King et al., 2014*) is predicted to abolish its enzymatic activity. The synthetic construct included GS repeat linkers and tagGFP2 codon optimized for mammalian cell expression. We inducibly tethered the nanocages to the plasma membrane using an N-terminal myristolation and palmitoylation motif and the FKBP/FRB* dimerization system, where FRB* is a T2098L variant of FRB that binds a rapamcyin analog, AP21967 (Clontech), and does not bind endogenous mTOR. These constructs bound weakly to the plasma membrane even in the absence of AP21967, presumably due to multivalent weak affinity binding. We transiently expressed these plasmid constructs into hiPS (or SK-MEL-2) cells with Lipofectamine Stem (Thermo Fisher). After 2 days we imaged the cells along with the genome edited cells using similar imaging settings.

## TIRF microscopy

Cells were imaged on either an Olympus IX-81 or Nikon Ti-2 inverted microscope fitted with TIRF optics. The IX-81 microscope used a 60 × 1.49 NA objective (Olympus) and an Orca Flash 4.0 sCMOS camera (Hamamatsu). Cells were illuminated with solid-state lasers (Melles Griot) with simultaneous acquisition by a DV-2 image splitter (MAG Biosystems). The microscope was maintained at 37 °C with a WeatherStation chamber and temperature controller (Precision Control) and images were acquired using Metamorph software. The Nikon Ti2 microscope was equipped with a motorized stage (Nikon), automated Z focus control, LU-N4 integrated four-wavelength solid state laser setup, TIRF illuminator (Nikon), quad wavelength filter cube, NI-DAQ triggering acquisition (National Instruments), an Orca Flash 4.0 sCMOS camera (Hamamatsu), and triggerable filter wheel (Finger Lakes Intstrumentation) with 525/50 and 600/50 wavelength emission filters. Cells were seeded on autoclaved 25 mm #1.5 round coverslips coated with 1 mL matrigel (80 µg/ mL) or recombinant Vitronectin-N diluted in PBS (Thermo Fisher). Cells were maintained at 37 °C with a stage top incubator (OKO Lab) and images were acquired with Nikon Elements.

## CK-666 experiments

CK-666 (Sigma) was reconstituted in DMSO and diluted in imaging media prior to treatment. Cells were treated prior to or during imaging. Cells were treated for 45 min prior to imaging unless otherwise indicated in the text. '0 µM' treatment corresponds to 0.1% DMSO treatment.

## Confocal microscopy

We imaged cells on a Nikon Eclipse Ti inverted microscope (Nikon Instruments) fitted with a CSU-X spinning disk confocal head (Yokogawa), four solid-state lasers (Nikon), iXon X3 EMCCD camera (Andor), and emission filter wheel (Sutter Instruments). The imaging area was kept at 37 °C with 5% $CO_2$ (In Vivo Scientific, LCI). We used a 100 × 1.45 NA Plan Apo oil immersion objective (Nikon). Images were acquired using Nikon Elements. We generally imaged 3–7 z slices with 300 nm z spacing at 3 s time intervals. Cells were seeded on sterile 4-chambered or 8-chambered #1.5H (0.170 ± 0.005 mm) cover glasses (CellVis). We imaged the cells in media supplemented with HEPES and the antioxidant oxyrase (OxyFluor) with substrate lactate. For quantitative fluorescence

experiments we limited the cells' exposure to ambient light, brightfield light and 488 nm laser light prior to acquisition, and used the same laser power during acquisition to compare experiments (generally 10% acousto-optic tunable filter (AOTF) power). For most experiments, we used DMEM/F12 without phenol red (Thermo Fisher) supplemented with the protein supplement used in StemFlex media (Thermo Fisher), which gave similar fluorescence intensity results as cells imaged in StemFlex (which has phenol red).

## Time-lapse imaging of nanocages
We treated cells with a range of concentrations of AP21967 for >45 min and then imaged them on a spinning disk confocal microscope at 0.3 s intervals for 1 min.

## Image correction
We took images of dilute GFP or autofluorescent Luria Broth (LB) to correct for uneven illumination in the imaging plane. The exposure time and laser power both scaled linearly on our instrument (*Figure 2—figure supplement 1*) which allowed us to adjust for dimmer or brighter signals by changing the exposure time.

## Cryo-electron tomography sample preparation
Holey carbon grids (Quantifoil R2/1, 200 mesh, gold) were glow discharged using a Pelco SC-6 sputter coater and sterilized in 70% ethanol in $H_2O$. SK-MEL-2 cells were plated in DMEM/F12 (Gibco) supplemented with 10% fetal bovine serum (premium grade, VWR Seradigm Life Science) and 1% Penicillin/Streptomycin (Gibco) onto grids and incubated overnight at 37°C and 5% $CO_2$ in a cell culture incubator. Samples were blotted and plunge frozen using a Vitrobot Mark IV (FEI) after addition of 10 nm BSA Gold Tracer (Electron Microscopy Sciences).

## Cryo-electron tomography data recording and processing
Tilt-series were recorded on a Titan Krios operated at 300 kV (FEI) equipped with a Quantum energy filter (Gatan) and a K2 direct electron detecting device (Gatan) at 2.97 Å pixel size and a target defocus of −2 μm. SerialEM (*Mastronarde, 2005*) in low-dose mode was used for automated tilt series acquisition using a bidirectional tilt scheme covering a whole tilt range from +60° to −60° with a base increment of 2° and a total electron dose of 100e⁻/Å². Tomograms were generated in IMOD (*Kremer et al., 1996*) using the gold beads as fiducials for tilt-series alignment. For *Figure 5*, tomograms were reconstructed using the simultaneous iterative reconstruction technique (SIRT) reconstruction algorithm, then filtered using the Nonlinear Anisotropic Diffusion (NAD) filter in IMOD and binned by a factor of 2. For *Figure 5—video 1*, we used the backprojection algorithm for tomogram reconstruction followed by a smoothing filter (Clip smooth function in IMOD). The U-shaped pit shown is one of six sites of CME we have identified in our tomograms, all of which have bent actin filaments around the pit to varying degrees. The bent filaments are especially prominent in the tomogram shown because of the orientation of the pit with respect to the missing wedge effect. A detailed treatment of these tomograms will be the subject of a subsequent study.

## Data analysis
*Calibration curve:* We used a combination of custom-written and publicly available image analysis software in Fiji (1.52i) and Matlab (r2017b) to analyze the traces of fluorescence intensity per spot, for multiple Z slices and time points. To measure fluorescence intensity per spot of GFP-tagged nanocages, we wrote a toolset in Fiji to select and circular regions of interest eight pixels (1.1 μm) in diameter, and used cross-correlation to center the regions around the intensity-based center of mass. We selected as background regions of concentric circles one pixel larger than the original region of interest. The toolset measured the fluorescence intensity per spot, which we subtracted by the area-corrected background intensity to yield fluorescence intensity per spot for the four constructs. Some analysis functions were adapted from published software (*Akamatsu et al., 2017*; *Epstein et al., 2018*; *McCormick et al., 2013*). We measured only spots that were contained within the slices imaged and were single stationary spots. For comparison of fluorescence to eGFP-MotB, we used smaller (6-pixel) ROIs.

We plotted background-subtracted fluorescence intensity per spot as a function of predicted copy number per structure to obtain a calibration curve relating fluorescence intensity per spot to numbers of molecules per structure. For the curve in *Figure 3D*, we combined data from three experiments with different imaging conditions by defining the average 60mer-GFP intensity per experiment as 1000 arbitrary units. Lines are linear fits through zero with $r^2$ calculated by linear least-squares fitting.

*Time-lapse fluorescence quantification:* We made modifications to automated MATLAB-based tracking software (*Aguet et al., 2013*; *Hong et al., 2015*) to track and analyze fluorescence-intensity time lapse data of genome-edited cells. The core tracking program (based on the software package µ-track) automatically identifies fluorescent spots and connects them as tracks by minimizing the linear assignment problem (*Jaqaman et al., 2008*). We used stringent tracking parameters with gap size 0 and search radius 0–2.3 pixels (248 nm). GFP and RFP tracks with high variability in the intensity/time profile were automatically rejected (*Ferguson et al., 2017*) as well as tracks $\leq$ 3 s in duration (*Dambournet et al., 2018*) and the remaining tracks were associated spatiotemporally according to a cost matrix (*Hong et al., 2015*). We used two track rejection schemes. In the first, users were presented with fluorescence montages and XY coordinates of the tracks to assess the fidelity of tracking for each event (*Hong et al., 2015*). In the second, tracks were automatically rejected based signal-to-noise ratio (>1.05) and proximity to neighboring tracks (>525 nm) (*Hong et al., 2015*). We checked that the manual and automatic track rejection schemes yielded similar results (lifetime distributions and intensity versus time plots) as well as to manual, kymograph-based quantification of lifetimes (below). From the above workflow (*Dambournet et al., 2018*) we increased throughput by connecting all steps into an automated tracking pipeline requiring minimal user input. For SK-MEL-2 cells expressing CLTA-RFP and DNM2-GFP, we tracked regions that were not near the nucleus (which has a concentration of Golgi-derived clathrin budding) and that did not have large, bright, persistent structures containing invariant RFP and GFP signals ('plaques,' which are likely sites of adhesion).

*Alignment method:* For *Figure 2G*, we aligned tracks based on the time point after the peak intensity in which 50% of the fluorescence remained in the GFP channel. We normalized the fluorescence intensity to compare movies from different imaging conditions. For *Figure 2I*, we aligned the tracks based on the maximum intensity. For *Figure 6D*, we aligned the tracks based on the disappearance of the RFP signal. This code is available at the following website: https://github.com/DrubinBarnes/Akamatsu_CME_manuscript.

*Manual track analysis:* We wrote a Fiji toolset that generated two-color kymographs from user-defined regions of interest, and then quantified the lifetime based on the lengths of the kymographs (from user-defined regions on the kymographs). This manual analysis was used in *Figure 6—figure supplement 1A* and for verification of the automated tracking scheme. This code is available at the following website: https://github.com/DrubinBarnes/Akamatsu_CME_manuscript.

*Nanocage particle tracking:* To track the membrane-tethered nanocages in 2D in cells we used TrackMate, a plugin available in Fiji that optimizes the Linear Assignment Problem (LAP) (*Jaqaman et al., 2008*). We detected spots using an estimate of 0.5 µm and threshold of 15, using a median filter and sub-pixel localization. We used the simple LAP tracker with a maximum linking and gap closing distance of 1 µm and two frames and a minimum track length of 4 frames.

*Calculating numbers of molecules per endocytic site:* We calculated the fluorescence intensity of background-subtracted ArpC3-GFP spots colocalized with AP2-RFP spots from single time-point images using the same background correction approach described for the calibration curve. We used the slope of the calibration curve to convert fluorescence intensity to numbers of molecules of ArpC3-GFP. Because the standard is inside cells, this standard controls for fluorescence environment and fluorescent protein folding and maturation. We used the resultant histogram of numbers of ArpC3-GFP per spot and the time-lapse fluorescence intensity data in *Figure 2G* to create the graph in *Figure 2I* of numbers of molecules of ArpC3-GFP over time.

## Acknowledgements

We would like to thank Dan Fletcher, Johannes Schoeneberg, and Jasmine Nirody for insightful comments on the manuscript; Julian Hassinger for advice on the continuum mechanics model and generating movies of the membrane simulations; Francois Nedelec for training and advice in Cytosim,

discussion of the relationship between association rate constants and binding rates, and sharing the 'Fork.cc' class for modeling Arp2/3 complex; Johannes Schoeneberg and Logan Akamatsu for help generating parallel simulation shell scripts; Sun Hong and Meiyan Jin for generating and validating the AP2-tagRFP-T human induced pluripotent cell line; and Elizabeth Li for contributions to automating the particle tracking pipeline. In addition, we would like to thank the UC Berkeley High Performance Computing cluster for training and server space for parallel simulations; the UC Berkeley QB3 MacroLab for purified *S. pyogenes* NLS-Cas9; the UC Berkeley Cancer Research Laboratory Molecular Imaging Center with support from the Gordon and Betty Moore Foundation; Karen M Davies and Jonathan Remis, Lawrence Berkeley National Labs Donner Building Cryo EM facility; Daniel B Toso and Paul Tobias, UC Berkeley Berkeley Bay Area Cryo-EM facility; NIH MIRA R35GM118149 to DGD; a postdoctoral fellowship from the Arnold and Mabel Beckman Foundation to MA; postdoctoral fellowship LT000234/2018L from the Human Frontier Science Program to DS; ARO W911NF1610411 and Office of naval research N00014-17-1-2628 to PR.

## Additional information

### Funding

| Funder | Grant reference number | Author |
|---|---|---|
| National Institutes of Health | R35GM118149 | David G Drubin |
| Arnold and Mabel Beckman Foundation | | Matthew Akamatsu |
| Human Frontier Science Program | LT000234/2018-L | Daniel Serwas |
| Army Research Office | W911NF1610411 | Padmini Rangamani |
| Office of Naval Research | N00014-17-1-2628 | Padmini Rangamani |

The funders had no role in study design, data collection and interpretation, or the decision to submit the work for publication.

### Author contributions

Matthew Akamatsu, Conceptualization, Software, Formal analysis, Funding acquisition, Investigation, Visualization, Methodology, Project administration; Ritvik Vasan, Michael A Ferrin, Software, Formal analysis, Investigation, Methodology; Daniel Serwas, Investigation, Methodology; Padmini Rangamani, Conceptualization, Supervision, Funding acquisition, Visualization, Project administration; David G Drubin, Conceptualization, Resources, Supervision, Funding acquisition, Visualization, Project administration

### Author ORCIDs

Matthew Akamatsu (iD) https://orcid.org/0000-0002-0286-5310
Daniel Serwas (iD) http://orcid.org/0000-0001-9010-7298
Michael A Ferrin (iD) https://orcid.org/0000-0002-9899-1169
Padmini Rangamani (iD) https://orcid.org/0000-0001-5953-4347
David G Drubin (iD) https://orcid.org/0000-0003-3002-6271

### Decision letter and Author response

Decision letter https://doi.org/10.7554/eLife.49840.sa1
Author response https://doi.org/10.7554/eLife.49840.sa2

## Additional files

### Supplementary files

- Supplementary file 1. Model parameters for continuum membrane mechanics model.
- Supplementary file 2. Parameters for coat pulling continuum mechanics simulations.

- Supplementary file 3. Parameters in the model.
- Transparent reporting form

## Data availability

All code associated with simulation and analysis is available at https://github.com/DrubinBarnes/Akamatsu_CME_manuscript (copy archived at https://github.com/elifesciences-publications/Akamatsu_CME_manuscript).

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
