## [Decision Letter]

**Acceptance summary:**

How actin polymerization generates forces during clathrin-mediated endocytosis in mammalian cells is still not clear. This work combines simulations of actin filaments and continuous modeling of membrane mechanics with high resolution experiments in living cells in order to study the self-organization of actin filaments during endocytosis. The number of Arp2/3 complexes (involved in branched actin network formation) has been measured in genome-edited human living cells and directly used in the simulations. The paper predicts that self-organization of actin filaments around the clathrin pit should produce enough force to pull on the bud in normal tension conditions, and to adapt to mechanical changes. It also predicts that some actin filaments should be bent, which was confirmed by cryoEM on intact cells, as well as how endocytosis should be quantitatively altered by molecular components involved in the process. Globally, this work should provide a solid basis for future quantitative studies to establish a molecular model of endocytosis.

**Decision letter after peer review:**

Thank you for submitting your article "Principles of self-organization and load adaptation by the actin cytoskeleton during clathrin-mediated endocytosis" for consideration by *eLife*. Your article has been reviewed by three peer reviewers, including Patricia Bassereau as the Reviewing Editor and Reviewer #1, and the evaluation has been overseen by Vivek Malhotra as the Senior Editor. The following individual involved in review of your submission has agreed to reveal their identity: Alex Mogilner (Reviewer #2).

The reviewers have discussed the reviews with one another and the Reviewing Editor has drafted this decision to help you prepare a revised submission.

Essential revisions:

We have some concerns that should be addressed or at least discussed in a revised version of the manuscript.

1) On a general ground, considering the modeling, the mechanical contribution to endocytosis of the membrane and of the rest of the system is included in an elastic force that opposes to the actin-driven deformation. However, in mammalian cells, budding happens very often independently of actin; many proteins are involved in budding before dynamin scission, producing elongation and bending forces on the membrane, thus favoring deformation, not the opposite. This is a key difference that should be justified at least.

2) Some initial assumptions about the initial localization of the NPF or Arp2/3, or the initiation of actin from small fragments on the coat should also be justified and the robustness of the model to these hypotheses should also be checked

3) The only direct testing of the prediction of the model is the observation of bent actin filaments around a bud. First, a clear evidence of the clathrin-related nature of the bud should be provided. Moreover, the simulations make many more predictions about the effect of certain parameters on internalization. It would reinforce the work if more critical tests of the model could be performed to confirm its validity, for instance, changing the density of actin nucleators, the capping rate (Figure 1—figure supplement 1) or the density of Hip1R (Figure 4) should have a strong effect, while changing the density of Arp2/3 should not (Figure 4—figure supplement 1).

[Editors' note: further revisions were suggested prior to acceptance, as described below.]

Thank you for resubmitting your work entitled "Principles of self-organization and load adaptation by the actin cytoskeleton during clathrin-mediated endocytosis" for further consideration by *eLife*. Your revised article has been evaluated by Vivek Malhotra as the Senior Editor, a Reviewing Editor and two reviewers.

The manuscript has been improved but there are some remaining issues that need to be addressed before acceptance, as outlined below:

*Reviewer #2*

1) Results subsection “Multiscale modeling shows that a minimal branched actin network is sufficient to internalize endocytic pits against physiological membrane tension “: '[…] direct scaling between resistance to internalization and membrane tension allowed us to treat this step of endocytic pit internalization as a linear spring […] ' – by internalization, do you mean some linear distance? Why does the force scale linearly with distance? (For example, force to push filopodia is constant, independent of the length…)

2) From the figures in which snapshots from the simulations are shown, it looks like an awful lot of filaments grows away from the pit, kind of contradicting the conclusions. Plates of the figures showing actin densities are hard to interpret. Basically, the authors need to think how to convey more convincingly the conclusion that the actin network is focused – at least, I could not convince myself that I see this from the simulation-generated data…

3) Readers need physical and geometric intuition about the results, and the text is a bit weak in that regard. When revising, pay attention at instances like 'simulations showed that…', and then the text goes to the next point – we need to understand qualitatively why the result is what it is.

*Reviewer #3:*

1) It would be very helpful to know the actual percentage of clathrin coated pits associated with bent actin filaments in their electron tomograms. It will help to grasp the actual contribution of actin filament bending on force generation.

2) It is still confusing when they talk about activated Arp2/3 complex, when they actually mean activated N-WASP. In essence, the Arp2/3 complex still attached to the actin branches is active too, and the authors do not really know where this pool of Arp2/3 is located. As they mention in the respond to the reviewer, what it is known in mammalian cells is the localization of N-WASP (presumably active) and not the position of the active Arp2/3 complex.

---

## [Author Response]

Essential revisions:We have some concerns that should be addressed or at least discussed in a revised version of the manuscript.1) On a general ground, considering the modeling, the mechanical contribution to endocytosis of the membrane and of the rest of the system is included in an elastic force that opposes to the actin-driven deformation.However, in mammalian cells, budding happens very often independently of actin; many proteins are involved in budding before dynamin scission, producing elongation and bending forces on the membrane, thus favoring deformation, not the opposite. This is a key difference that should be justified at least.

We thank the reviewers for giving us the opportunity to summarize the available data of actin’s participation in endocytic membrane deformation in cooperation with other endocytic proteins. We think it is critically important for the field to have an open and robust discussion of actin’s role in mammalian endocytosis based on the available data. We have modified the Introduction and Results to better distinguish the contributions of actin from other curvature-generating endocytic proteins.

There is an energetic cost to deform a flat plasma membrane into a spherical vesicle based on the membrane’s rigidity and tension (Hassinger et al., 2017 and the present study). The magnitude of this energy requirement depends on the strength of the membrane’s rigidity and tension (Hassinger et al., 2017, and the present study). In yeast, high turgor pressure further increases this energetic cost (Dmitrieff et al., 2015). As the reviewers note, many endocytic proteins participate in the deformation of the plasma membrane during endocytosis, particularly coat proteins that sense and generate membrane curvature (Stachowiak et al., 2012). Actin is one such protein that helps to overcome this energy barrier.

When the energy barrier to deformation is especially high, coat proteins are not sufficient to complete endocytosis. The present study indicates that the most energetically costly step in mammalian CME is the U to omega shape transition. Most cells experience variations in membrane tension – think of the stretch we put on our tissues every day – which is likely to vary the energetic requirement for membrane deformation and continue to provide cases in which the coat proteins alone are not sufficient to advance endocytosis.

Independence of actin:

As the reviewers note, there are cases in which endocytosis appears to proceed independent of actin assembly. This observation is consistent with the above arguments that in mechanically permissive conditions, endocytic coat proteins are sufficient to carry out endocytosis. The percentage of CME events reported to colocalize with actin ranges from 50% to 85%, depending on cell type, imaging modality, and fluorescent marker of the actin cytoskeleton (Grassart et al., 2014; Li et al., 2015; Yoshida et al., 2018; present study). Effects of inhibiting actin dynamics with small molecules are difficult to interpret in animal cells because these treatments often disrupt the actin cortex, which contributes to overall membrane tension and therefore resistance to internalization (Pontes et al., 2017). Therefore, actin inhibitor small molecules have opposing effects on CME in animal cells. Additionally, these treatments do not completely remove endocytic actin filaments (Collins et al., 2011).

Role of polymerized actin in mammalian clathrin-mediated endocytosis:

In the presence of small molecule compounds that depolymerize actin filaments (Latrunculin A) or promote nucleation and prevent depolymerization of actin filaments (Jasplakinolide), more endocytic pits stall at the U or invaginated intermediate shapes, particularly at the apical surface of cells (Yarar et al., 2005, Boulant et al., 2011). Functional actin and Arp2/3 complex is required for the efficient closure of endocytic pits (Yoshida et al., 2018). Latrunculin A or Arp2/3 complex inhibitor CK666 stalls clathrin-mediated endocytosis (Grassart et al., 2014; current study). Latrunculin A and Jasplakinolide stall clathrin-mediated endocytosis when membrane tension is high (apical surface of cells or stretched cells) (Boulant et al., 2011).

Other proteins favor membrane deformation:

The reviewers bring up an important point that other endocytic proteins favor rather than inhibit membrane deformation. In our membrane mechanics model, this is modeled as an endocytic coat with increased rigidity and spontaneous (preferred) curvature relative to the rest of the plasma membrane. This modeling (Hassinger et al., 2017; Figure 1B-D) includes such contributions from endocytic coat proteins. Indeed, the spontaneous curvature of the coat also assists actin in advancing the pit from a U shaped intermediate to the late-stage omega shape. Similarly, in our Cytosim model, we initialize the endocytic pit as a hemisphere, to model endocytic sites deformed into a U shape by the other endocytic proteins participating in budding, but stalled at the U shape under elevated tension in the absence of functional actin. Therefore the resistance to membrane deformation mostly arises from the mechanical properties of the membrane itself. In our Cytosim model we focused on the mechanism by which actin contributes to endocytosis because it is poorly understood and because this system provides an opportunity to establish generalizable principles for how actin forces act on membranes. We modified the Results subsection “Multiscale modeling shows that a minimal branched actin network is sufficient to internalize endocytic pits against physiological membrane tension” to emphasize that, in both models, endocytic coat proteins contribute to rather than inhibit membrane deformation.

2) Some initial assumptions about the initial localization of the NPF or Arp2/3, or the initiation of actin from small fragments on the coat should also be justified and the robustness of the model to these hypotheses should also be checked.

We thank the reviewers for these two suggestions to more rigorously explore our assumptions about the initial conditions in the model. To address these concerns we ran additional simulations which are summarized below and incorporated into the revised manuscript.

Position of active Arp2/3 complex:

We assume that pre-activated Arp2/3 complex resides in a ring around the base of the endocytic pit. This assumption is based on the following observations of activators of Arp2/3 complex. In budding yeast, the N-WASP homologue Las17 resides in a ring around the base of the endocytic pit (Mund et al., 2018). In mammalian cells, the N-WASP activator FCHSD2 similarly resides in a ring around the base of the pit (Almeida-Souiza, 2017). To simplify the model and limit assumptions that are not constrained by experimental data, we treat the coincidence of N-WASP and Arp2/3 complex as pre-activated Arp2/3 complex in a ring around the base of the pit. We added a description explaining this assumption in the Results section.

Later in endocytosis, the N-WASP activator SNX9 localizes to a ring around the neck of the pit (Schoeneberg et al., 2017). Our model focuses on the U-to-omega shape transition in endocytosis, so this later stage of endocytosis is not well-captured in our model. Furthermore, to ascertain how the position of active Arp2/3 complex might affect the results of our simulations, we conducted additional simulations with 50 of the 200 pre-activated Arp2/3 complex localized in an additional collar near the neck that is revealed later in endocytosis (Figure 4—figure supplement 2). These simulations show that in our model, the presence of additional Arp2/3 complex near the neck of the pit did not appreciably impact the extent of internalization or the number of actin filament barbed ends near the neck of the pit.

Position of mother filaments:

We apologize that the original version of the manuscript only described one initial arrangement of mother filaments, which are necessary to template the nucleation of new branched actin filaments via N-WASP and the Arp2/3 complex. The origin of these mother filaments in animal cells is not currently known. In this model, we assumed that linear actin filaments concentrate near the site of endocytosis to trigger branched actin assembly. We tested the impact of this assumption in a series of additional simulations. Based on the following data below, which we have added to new Figure 1—figure supplement 2, this assumption improved the reproducibility of the timing of the simulations, but otherwise did not affect the outcome detectably.

We initially used a more general assumption for the location of mother filaments, namely that a cytoplasmic pool of short linear actin filaments seeds endocytic actin assembly. The cytoplasm of mammalian cells contains ~5 µM of short actin filaments, with an average length of 35 nm and diffusion coefficient of 5 µm^2^/s (Raz-Ben Aroush et al., 2017). Such cytoplasmic linear actin filaments are predicted to be important for initiation of endocytosis in fission yeast (Chen et al., 2013). In our model, this distribution of actin filaments also triggered the internalization of endocytic pits. However, the timing of initiation varied widely (new Figure 1—figure supplement 2).

To capture the full endocytic event in our simulations we instead seeded a limited number of randomly oriented actin filaments near the endocytic pit. Cortical actin filaments may also help to template the assembly of actin filaments, but their precise locations are not currently known.

To complement simulations varying the number of nucleators (which determine the number of mother filaments) (Figure 1—figure supplement 1F), we ran additional simulations in which we varied the surface density of 30 nucleators in a ring around the base of the pit.

In these simulations, endocytic internalization was not sensitive to the surface density of nucleator proteins, which determines the initial distribution of mother filaments. This probably occurs due to the diffusion rate of mother filaments and the lack of attachments to push against. These data are now in Figure 4—figure supplement 2. The strategies the cell uses to concentrate mother filaments for timely actin assembly will be subject of future studies.

3) The only direct testing of the prediction of the model is the observation of bent actin filaments around a bud. First, a clear evidence of the clathrin-related nature of the bud should be provided.

We appreciate this comment and regret not doing a better job of showing how we identified this site as clathrin-related. To demonstrate that the invagination in the tomogram is a clathrin-coated pit, we now highlighted the positions of a lattice of clathrin hexagons and pentagons around a section of the invagination in the tomogram (Figure 5—figure supplement 1). Moreover, the simulations make many more predictions about the effect of certain parameters on internalization. It would reinforce the work if more critical tests of the model could be performed to confirm its validity, for instance, changing the density of actin nucleators, the capping rate (Figure 1—figure supplement 1) or the density of Hip1R (Figure 4) should have a strong effect, while changing the density of Arp2/3 should not (Figure 4—figure supplement 1).

Additional tests of the model:

We thank the reviewers for the opportunity to identify additional experimental data consistent with the model to confirm its validity. In the revised manuscript, we have added a figure summarizing validations of the model from published literature as well as predictions of the model tested by our experimental data (Figure 8—figure supplement 1). These points are now emphasized in a new paragraph in the Discussion.

1) Hip1R knockdown: Previously, our lab knocked down Hip1R in mammalian cells and observed a resultant inhibition of endocytosis (Engqvist-Goldsten et al., 2004). With strong (>90%) knockdown of Hip1R, endocytic internalization (assayed by transferrin uptake) was inhibited by up to 80%. This result is consistent with our simulations showing that a threshold number of actin linkers (such as Hip1R) are necessary for endocytic internalization (Figure 4—figure supplement 3A). This comparison in new Figure 8—figure supplement 1A-B.

2) Capping rate: Actin filaments can be capped with the small molecule compound Cytochalasin, which blocks the barbed ends of actin filaments. Indeed, treatment of *Dictyostelium* or SK-MEL-2 cells with Cytochalasin A or D inhibits CME, assayed by an increase in stalled endocytic sites marked by clathrin-RFP (Brady et al., 2010) or the slower accumulation of dynamin2-GFP at endocytic sites (Grassart et al., 2014). This effect was observable even though endocytic actin filaments often persist under Cytochalasin and Latrunculin treatment (Collins et al., 2011). This comparison is in new Figure 8—figure supplement 1C-E.

3) The presence of bent filaments at CCPs, as noted by the reviewers (Figure 8—figure supplement 1F-G).

4) CK666 experiments: We treated cells with the Arp2/3 complex inhibitor CK-666 and found a dose-dependent inhibition of endocytosis, assayed by the lifetime of endogenously tagged clathrin and dynamin (Figure 6D). This result is consistent with simulations showing that endocytic internalization depends on the activity (nucleation rate) of Arp2/3 complex (Figure 6C). Moreover, the simulations predict that at physiological membrane tension in SK-MEL2 cells (0.12 pN/nm), the basal nucleation rate of Arp2/3 complex (1 filament per second) is the minimum rate required for strong internalization (Figures 6B and 6C and new Figure 8—figure supplement 1H-I).

We realized that this important result was buried in a complicated figure in which both membrane tension and Arp2/3 complex activity were varied. Therefore, we split this figure into two. The first figure (Figure 6) details the modeling and experimental results on the effect of Arp2/3 complex activity on endocytosis. The second figure (new Figure 8) extends these results to a variety of values of membrane tension, which set up important predictions for future experimental studies.

5) Load adaptation of the endocytic actin network: Our simulations predict that in response to increased membrane tension, the actin network adapts by binding more active Arp2/3 complex (Figure 7G), thereby nucleating more actin filaments. Load adaptation by the actin network is also the subject of the Kaplan et al. manuscript, which we have provided to the reviewers. Indeed, Kaplan et al. found that when membrane tension was increased, more actin grew around sites of clathrin-mediated endocytosis in mammalian cells. A more detailed comparison to those experimental data will be the subject of a future study.

We anticipate that additional tests of the model will be the subject of future studies.

[Editors' note: further revisions were suggested prior to acceptance, as described below.]

The manuscript has been improved but there are some remaining issues that need to be addressed before acceptance, as outlined below:Reviewer #21) Results subsection “Multiscale modeling shows that a minimal branched actin network is sufficient to internalize endocytic pits against physiological membrane tension “: '[…] direct scaling between resistance to internalization and membrane tension allowed us to treat this step of endocytic pit internalization as a linear spring' – by internalization, do you mean some linear distance?

As clarified in the revised text, internalization refers to the linear distance that the head of endocytic pit moves inwards (-Z). We used the term “internalization” because we deemed it to be more intuitive to our target audience than the more technical term of displacement.

We have added the following sentences to the Results and the Materials and methods sections to make it clear that we define internalization as the displacement in Z:

“Here, extension refers to pit internalization, which is a displacement in the -Z direction (Figure 1A-B).”

“Internalization is defined as a displacement in the -Z direction (Figure 1A).”

Why does the force scale linearly with distance? (For example, force to push filopodia is constant, independent of the length…)

In Figure 1C, the linear scaling of force with distance is a classic result in membrane mechanics in the linear regime. This was shown in the classic membrane mechanics paper of Derenyi et al., 2002, and more recently in Alimohamadi et al., 2018. There are two regimes of force-displacement relationships for deforming a membrane. The early stage shows a linear relationship between the force and displacement for bending the membrane. Once a critical force threshold is achieved, the force required to increase the displacement is constant. The reviewer, when invoking the example of the constant force required to push a filopodium, is referring to the second phase where further membrane bending is not required. We are focused on the regime of membrane bending to form an invagination.

In response to the reviewer’s comments, we have added a new paragraph to the Materials and methods section, which more extensively describes the relationship between force and extension of bare membranes:

Subsection “Membrane mechanics module”: “Continuum mechanics modeling of the plasma membrane allows a quantitative understanding of the relationship between applied forces and the shape of the membrane (Derenyi et al., 2002; Rangamani et al., 2013). […] Forces due to actin polymerization can also help overcome the energy barrier (Hassinger et al., 2017), but the relationship between applied actin forces and coated membrane shape has not been explored quantitatively.”

2) From the figures in which snapshots from the simulations are shown, it looks like an awful lot of filaments grows away from the pit, kind of contradicting the conclusions. Plates of the figures showing actin densities are hard to interpret. Basically, the authors need to think how to convey more convincingly the conclusion that the actin network is focused – at least, I could not convince myself that I see this from the simulation-generated data…

We thank the reviewer for an opportunity to provide additional evidence supporting the mechanism of self-organization identified in our simulations, and to distinguish it from the organization of lamellipodial actin filaments.

In response to the reviewer’s comment we have added four new panels to Figure 4—figure supplement 3 which show (1) the initial orientation of actin filaments in the simulations; (2) a schematic that illustrates the mechanism of self-organization; and (3) plots of the evolution of the axial orientation of the filaments over time, relative to that of randomly oriented filaments. We also narrowed the focus of the Results section to emphasize that the filaments near the base of the pit, the productive filaments, are the ones that exhibit this self-organization.

Subsection “Self-organization of actin filaments into a radial dendritic network drives endocytic internalization”: “Based on the initial geometry of the endocytic pit and activated Arp2/3 complex, branched actin filaments self-organized into a radial dendritic network: the network attached to the clathrin coat by binding to Hip1R, the pointed (minus) ends localized close to the pit and the barbed (plus) ends near the base of the pit were oriented to grow toward the base of the pit.”

3) Readers need physical and geometric intuition about the results, and the text is a bit weak in that regard. When revising, pay attention at instances like 'simulations showed that…', and then the text goes to the next point – we need to understand qualitatively why the result is what it is.

We thank the reviewer for the opportunity to better explain the mechanistic results in our manuscript based on physical and geometric intuition. In response, we elaborated on the following points in the Results and Discussion sections:

Subsection “Spatial distribution of actin/coat attachments and Arp2/3 complex, but not Arp2/3 complex density, strongly affects actin self-organization and pit internalization”: “Filaments growing toward the base of the pit encounter active Arp2/3 complex, which catalyzes dendritic nucleation of new actin filaments growing in a similar direction (Figure 4—figure supplement 3B and D; (Carlsson, 2001). As a result, near the base of the pit, filaments increasingly orient toward the base of the pit over time.”

Discussion: “Our simulations showed that a threshold number of actin linkers such as Hip1R is necessary for endocytic internalization (Figure 4—figure supplement 1B and Figure 8—figure supplement 1A-B). This threshold appears necessary to allow efficient filament capture by the coat and force transmission from the actin network to the coat.”

Discussion: “We also showed in our model that capping rate is an important parameter for progression of CME; our simulations show that increasing capping rate of actin filaments inhibits CME, presumably because increasing capping decreases the total amount of actin.”

Discussion: “In this study, our simulations predicted that actin filaments bend around endocytic pits. These bent filaments store elastic energy for subsequent force production much as a pole vaulter’s pole bends and stores energy for delayed force production.”

Discussion: “Without sufficient Arp2/3 complex, CME fails due to insufficient force production.”

We hope that these additional explanations make the manuscript read as more physically intuitive.

Reviewer #3:1) It would be very helpful to know the actual percentage of clathrin coated pits associated with bent actin filaments in their electron tomograms. It will help to grasp the actual contribution of actin filament bending on force generation.

We thank the reviewer for this important comment. We have identified six clathrin-coated pits in our tomograms, and every pit (100%) associated with bent actin filaments. However, this observation does not reveal whether the filaments could have been bent by thermal fluctuations alone or were actively under load. Quantitative analysis of these tomograms will be the subject of an ongoing study in preparation. In response to the reviewer’s comment we added the following text to the Materials and methods section:

Subsection “Cryo-electron tomography data recording and processing”: “The U-shaped pit shown is one of six sites of CME we have identified in our tomograms, all of which varying amounts of have bent actin filaments around the pit to varying degrees. […] A detailed treatment of these tomograms will be the subject of a subsequent study.”

2) It is still confusing when they talk about activated Arp2/3 complex, when they actually mean activated N-WASP. In essence, the Arp2/3 complex still attached to the actin branches is active too, and the authors do not really know where this pool of Arp2/3 is located. As they mention in the respond to the reviewer, what it is known in mammalian cells is the localization of N-WASP (presumably active) and not the position of the active Arp2/3 complex.

We apologize that we had not made the text clearer to delineate active N-WASP from active Arp2/3 complex. In response to the reviewer’s comment, we extensively rewrote the Materials and methods section to better distinguish active N-WASP from active Arp2/3 complex in the setup of our model:

Subsection “Active Arp2/3 complex”: “We collapsed the activation steps of Arp2/3 complex into a single species, active Arp2/3 complex, that resides on the plasma membrane from the beginning of the simulation. This models the cellular process, in which soluble Arp2/3 complex is inactive until it encounters its activator N-WASP at the plasma membrane.”

Subsection “Active Arp2/3 complex”: “Because the activation rate and concentrations of these proteins are not yet known, we considered fully active N-WASP (similar to the VCA region alone) rather than modeling the individual activation steps as the relevant species to choose parameters. […] Thus, this model aims to functionally capture Arp2/3 complex activation and the geometry of branched actin filament nucleation, rather than explicitly modeling each molecule involved in the process of Arp2/3 complex activation.”

The above text is based on the information about the multistep activation of Arp2/3 complex, for example schematized in Espinoza Sanchez, S., Metskas, L.A., Chou, S.Z., Rhoades, E., and Pollard, T.D. PNAS (2018).

In future modeling efforts, we hope to be able to capture this critical multistep activation mechanism in more explicit molecular detail. However, currently there is not enough quantitative information available to constrain a model containing all of those steps, particularly in metazoan cells. Therefore, our model focuses on the functional steps necessary to make a new branched actin filament. Since a new branch only occurs in the coincidence of active N-WASP, Arp2/3 complex, and a mother filament, based on the position of N-WASP activators, this occurs in a ring around the base of the pit. As such, in our model, active Arp2/3 complex only nucleates a new branch when active Arp2/3 complex at the base of the pit – functionally equivalent to the coincidence of Arp2/3 complex and active N-WASP – comes in proximity to an existing actin filament (the mother filament). This is why we believe that it is reasonable to simplify this activation step as a pre-activated Arp2/3 complex that only nucleates a filament in association with a mother filament.